# RETHINKING THE STABILITY-PLASTICITY TRADE-OFF IN CONTINUAL LEARNING FROM AN ARCHITECTURAL PERSPECTIVE

## ABSTRACT

The quest for Continual Learning (CL) seeks to empower neural networks with the ability to learn and adapt incrementally. Central to this pursuit is addressing the stability-plasticity dilemma, which involves striking a balance between two conflicting objectives: preserving previously learned knowledge and acquiring new knowledge. Existing studies have proposed numerous CL methods to achieve this trade-off. However, these methods often overlook the impact of basic architecture on stability and plasticity, thus the trade-off is limited to the parameter level. In this paper, we delve into the conflict between stability and plasticity at the architectural level. We reveal that under an equal parameter constraint, deeper networks exhibit better plasticity, while wider networks are characterized by superior stability. To address this architectural-level dilemma, we introduce a novel framework denoted Dual-Architecture (Dual-Arch), which serves as a plug-in component for CL. This framework leverages the complementary strengths of two distinct and independent networks: one dedicated to plasticity and the other to stability. Each network is designed with a specialized and lightweight architecture, tailored to its respective objective. Extensive experiments across datasets and CL methods demonstrate that Dual-Arch can enhance the performance of existing CL methods while being up to **87%** more compact in terms of parameters than the baselines.

## 1 INTRODUCTION

Continual Learning (CL) seeks to enable neural networks to continuously acquire and update knowledge. The primary challenge in CL is catastrophic forgetting McCloskey & Cohen (1989); Goodfellow et al. (2013), i.e., directly updating neural networks to learn new data causes rapid forgetting of previously acquired knowledge. To learn continually without forgetting, a neural network must balance plasticity, to learn new concepts, and stability, to retain acquired knowledge. However, emphasizing stability can limit the neural network's ability to acquire new knowledge, while excessive plasticity can lead to severe forgetting, a challenge known as the stability-plasticity dilemma Grossberg (2013).

To enhance CL, most of the research efforts Li & Hoiem (2017); Henning et al. (2021); Feng et al. (2022) are centered on developing novel learning methods that achieve a better trade-off between stability and plasticity. These methods involve adding loss terms that prevent the model from changing, replaying past data, or explicitly using distinct parts of the network for different tasks, *etc* Wang et al. (2023b). In particular, architecture-based methods have achieved great success across various CL scenarios Rusu et al. (2016); Rosenfeld & Tsotsos (2018); Wang et al. (2023a). Characteristically, this type of method introduces an extra part of the network that is solely trained on the current data, which is then integrated with other parts that have been continuously trained on the previous data Yan et al. (2021); Zhou et al. (2023b). Since a new independent parameter space is used to learn the current data, these methods avoid rewriting the original parameters, thus preserving the old knowledge. In this way, the conflict between stability and plasticity at the parameter level can be significantly mitigated.

While studies that focus on expanding and allocating architecture have achieved notable success, research on the basic architectures for CL is still in its infancy. This gap is crucial because, despite the ability of advanced learning methods to optimize parameters effectively, the overall CL performance remains constrained by suboptimal architectures Lu et al. (2024). In this regard, certain pioneer

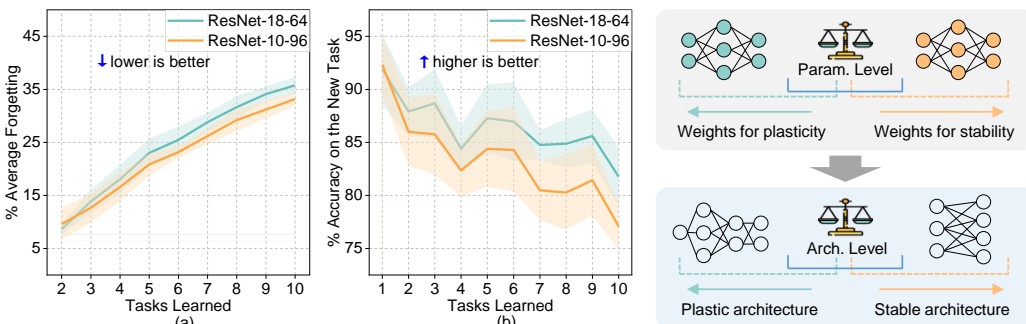

Figure 1: **Left.** (a) The average forgetting and (b) the accuracy on the new task of ResNet-18 and its wider and shallower variant. Details are presented in Sec. 3. **Right.** While existing research mainly optimizes weights (represented by node colors) for the stability-plasticity trade-off at the parameter level, this study proposes a novel insight for extending this trade-off to the architectural level.

works have concluded that wider and shallower networks exhibit superior overall CL performance, mainly contributing to enhanced stability Mirzadeh et al. (2022a;b). However, theoretical analyses and practices Simonyan & Zisserman (2014); He et al. (2016); Liang & Srikant (2016); Raghu et al. (2017) have demonstrated that deeper networks possess enhanced representation learning ability, indicating the important role of depth in facilitating plasticity. These findings raise a concern regarding whether there is an inherent conflict between stability and plasticity at the architectural level under a given parameter count constraint.

To investigate this, we conducted a comparison between ResNet-18 He et al. (2016) and its wider yet shallower variant, evaluating their average forgetting and accuracy on the new task. As shown in Fig. 1, ResNet-18 achieves higher accuracy on the new task, indicative of better plasticity, whereas the wider yet shallower variant exhibits lower average forgetting, indicative of greater stability. However, both networks underperform in the other aspect, which indicates there may exist a stability-plasticity dilemma at the architectural level as well. Given that existing works Zhou et al. (2023b); Lu et al. (2024) typically employ a uniform architecture for both stability and plasticity, this inherent dilemma may limit CL performance, even when the architecture and parameters are finely optimized.

*How to balance the stability and plasticity at the architectural level?* An intuitive and straightforward solution is to combine two independent models with distinct architectures: one dedicated to plasticity and the other to stability. Previous studies on CL have demonstrated that incorporating an auxiliary model, specifically trained on the current data, can enhance the plasticity of the primary model Kim et al. (2023); Bonato et al. (2024). Building on these insights, we extend from an architectural perspective, proposing a novel framework that employs a plastic architecture to acquire new knowledge, which is then transferred to the main model with a stable architecture. Specifically, knowledge distillation Hinton et al. (2015); Romero et al. (2014) is utilized for this transfer due to its proven efficacy in transferring knowledge between networks with different architectures Gou et al. (2021). Consequently, our proposed framework, **Dual-Arch**itecture (Dual-Arch), leverages the complementary strengths of two distinct architectures, effectively balancing stability and plasticity at the architectural level. Extensive experiments show that Dual-Arch markedly enhances CL performance with significantly fewer parameters when compared to the baselines. Code is available at `https://github.com/anonymous-dual-arch/d-arch`.

The contributions of this study are outlined as follows:

- Through meticulous empirical studies, we demonstrate that existing architectural designs typically exhibit good plasticity but poor stability, while their wider and shallower variants exhibit the opposite traits. Based on these findings, we propose a novel insight for exploring the stability-plasticity trade-off from an architectural perspective.

- We introduce a novel CL framework, Dual-Arch, which employs dual architectures dedicated to stability and plasticity and thus combines both advantages. Furthermore, Dual-Arch can be naturally incorporated with various CL methods as a plug-and-play component.

- Extensive experiments demonstrate that Dual-Arch is parameter-efficient, i.e., attaining better performance with a remarkably reduced parameter count than using a single architecture.

## 2 RELATED WORK

CL involves letting models sequentially learn a series of tasks without or with limited access to previous tasks. Typically, these tasks are framed as visual classification problems Masana et al. (2022). Based on whether the task identity is provided or must be inferred, CL can be categorized into three typical scenarios: Task/Class/Domain Incremental Learning (IL) Van de Ven et al. (2022). Moreover, there are some works that focus on more challenging scenarios where the task boundaries are blurry Arani et al. (2022); Sarfraz et al. (2022). In the *Task IL* scenario, the task identity of test samples is accessible at inference time, so the networks only need to learn and remember how to classify within each task. More generally, the *Class IL* scenario requires networks to predict both the task identity and the sub-class label. Moreover, the *Domain IL* scenario does not introduce new classes during the learning process, instead, it characterizes task changes by introducing a shift in the input distribution. This study mainly focuses on the *Class IL* scenario, which is regarded as the most general and realistic among three typical scenarios Van de Ven et al. (2022); Wang et al. (2023b).

### 2.1 LEARNING METHODS FOR CL

To address catastrophic forgetting, the CL community has developed numerous CL methods aimed at striking a balance between stability and plasticity. These methods encompass a range of techniques, including memory replay, weight or function regularization, and dynamic architecture.

**Replay-based methods** keep a subset of the previous data information in a memory buffer and thus exploit it to recover old data distributions. A straightforward implementation involves simply replaying the stored data in conjunction with the current data (i.e., joint training), thereby simulating the training process on a dataset that is independently and identically distributed Robins (1995); Chaudhry et al. (2018). It should be noted that such a strategy is particularly widely used in *Class IL* and often combined with other categories of techniques Wang et al. (2023b). However, these methods require access to raw past data, which might be discouraged in some environments due to privacy concerns. Instead, recently some works Lin et al. (2022); Sun et al. (2023); Lin et al. (2023) elaborately construct a special parameter space of old tasks as the memory.

**Regularization-based methods** incorporate a regularization loss term to balance old and new tasks, which can be divided into two sub-directions based on the regularization target Wang et al. (2023b). The first is *weight regularization*, which selectively constrains the variation of the network parameters based on the importance of each parameter in performing the old tasks, e.g., EWC Kirkpatrick et al. (2017) and SI Zenke et al. (2017). The second is *function regularization*, which targets the intermediate or final output of the prediction function. This strategy typically involves transferring knowledge from previous CL models to the current model through knowledge distillation to mitigate forgetting Madaan et al. (2023). For instance, LwF Li & Hoiem (2017) proposes to let the model concurrently learn the soft target generated by the previous model alongside the new data. Furthermore, several methods integrate knowledge distillation with memory replay, proposing more advanced solutions for *Class IL*, such as iCaRL Rebuffi et al. (2017) and WA Zhao et al. (2020).

**Architecture-based methods** mitigate inter-task interference and thus balance stability and plasticity by allocating an expanding incremental parameter space of the network for each new task Yoon et al. (2017). DER Yan et al. (2021) and MEMO Zhou et al. (2023b) are two typical representatives of this type of method, both of which incorporate memory replay. While these methods have demonstrated impressive CL performance, the rapid growth of parameters presents a challenge Li et al. (2019); Zhou et al. (2023b). This may limit their application, especially in memory-restricted scenarios.

**Discussion.** In principle, the performance of neural networks is jointly influenced by their parameters and architectures. While the learning methods mentioned above mainly enhance CL by optimizing the parameters or extending parameter space, the suboptimal basic architectures might still limit CL performance. Our study aims to address this by proposing a plug-and-play framework that leverages the complementary strengths of two distinct architectures.

### 2.2 NEURAL ARCHITECTURES FOR CL

Besides learning methods, there is a body of research Mirzadeh et al. (2022a;b); Pham et al. (2022) that concentrates on exploring optimal neural architectures for CL. In particular, ArchCraft Lu et al.

(2024) delves into the influence of various network components and scaling on CL performance, demonstrating that certain architectural designs are more CL-friendly than existing ones. Furthermore, it is shown that a well-designed architecture can achieve superior CL performance with a smaller parameter count, which is particularly beneficial for memory-constrained environments Lu et al. (2024). These studies emphasize that the impact of architectural designs on CL performance is at least as significant as that of the learning methods. However, it should be noted that existing studies focus exclusively on the impact of architectures on the overall performance of CL. Our work extends this line of inquiry by highlighting the inherent conflict between stability and plasticity at the architectural level and subsequently proposing a novel solution to address it.

### 2.3 MULTI MODELS FOR CL

Various existing studies have proposed employing additional models to enhance CL Li & Hoiem (2017); Kim et al. (2023); Bonato et al. (2024). In particular, certain works Pham et al. (2021); Arani et al. (2022) based on complementary learning systems McClelland et al. (1995); Kumaran et al. (2016) utilize two learners (known as slow and fast learners) with different functions to achieve CL. Our proposed solution shares a similar conceptual framework, crafting two independent learners that assume roles of plasticity and stability respectively during the CL process. However, unlike these prior efforts that employ a uniform architecture for all models, our study emphasizes the importance of specific architectural designs tailored to each learner. By doing so, our study provides novel insights into more effectively leveraging multiple models for CL.

## 3 ARCHITECTURAL DIMENSIONS OF STABILITY AND PLASTICITY

This section presents an investigation of the impact of architectural designs on the stability and plasticity of neural networks. The primary objective of this investigation is to reveal the conflict between stability and plasticity at the architectural level, with a focus on network scaling.

### 3.1 EVALUATION SETTINGS

**Architectural Variants.** ResNet-18 He et al. (2016) is selected as the foundational architecture, given its extensive utilization in existing CL research Yan et al. (2021); Goswami et al. (2024). Our primary focus is on examining the impact of depth and width on CL. To this end, we vary the number of layers and initial channel counts in ResNet-18 while maintaining a relatively constant total parameter count. Additionally, we conduct an extended study to investigate the effect of pre-classification width. This involves replacing the global average pooling (GAP) layer with a $4 \times 4$ average pooling layer with a stride of 3, thereby producing an output feature map of size $2 \times 2$.

**Implementation Setup.** A subset of ImageNet Deng et al. (2009), known as ImageNet100 Rebuffi et al. (2017), is utilized as the dataset, and it is partitioned into 10 incremental tasks, each comprising 10 classes. All models are trained using iCaRL Rebuffi et al. (2017), a classic learning method in the CL field, with a fixed memory size of 2,000 exemplars.

**Evaluation Metrics.** To assess the plasticity, we measure the Average Accuracy on the New task (AAN) across all incremental steps. A higher AAN value signifies greater plasticity. Furthermore, we utilize the Average Forgetting (AF) metric to evaluate the stability, with lower AF values indicating superior stability. Specifically, the AF after learning the $k$-th task is defined as:

$$AF_k = \frac{1}{k-1} \sum_{b=1}^{k-1} (a_b^* - a_b), \tag{1}$$

where $a_b$ denotes the current performance of task $b$, and $a_b^*$ represents its maximum performance in the past. In particular, we use the AF after learning the Final task (FAF) as the overall stability metric.

### 3.2 EVALUATION RESULTS

The performance comparison between ResNet-18 and its variants, under comparable parameter counts (within a $\pm 3\%$ margin), is summarized in Tab. 1. It can be observed that the wider yet shallower variant demonstrates decreases in AAN by 2.97% (83.44% vs. 86.41%) and FAF by 2.60%

Table 1: The AAN and FAF (%) of the original ResNet-18 (gray background) and its variants. We report the mean and std of 5 runs with different task orders. Note that the '#P' denotes the parameter counts of a single architecture here.

| Depth | Width | Penultimate Layer | #P (M) | AAN ↑ | FAF ↓ |
|---|---|---|---|---|---|
| 18 | 64 | GAP | 11.23 | 86.41±0.60 | 35.76±1.62 |
| 10 | 96 | GAP | 11.10 | 83.44±0.84 (-2.97) | 33.16±1.28 (-2.60) |
| 18 | 64 | $4 \times 4$ AvgPool | 11.38 | 84.64±0.43 (-1.77) | 34.17±2.03 (-1.59) |
| 26 | 52 | GAP | 11.56 | 86.68±0.70 (+0.27) | 36.02±1.79 (+0.26) |
| 34 | 46 | GAP | 11.04 | 86.87±0.54 (+0.43) | 35.98±1.97 (+0.22) |

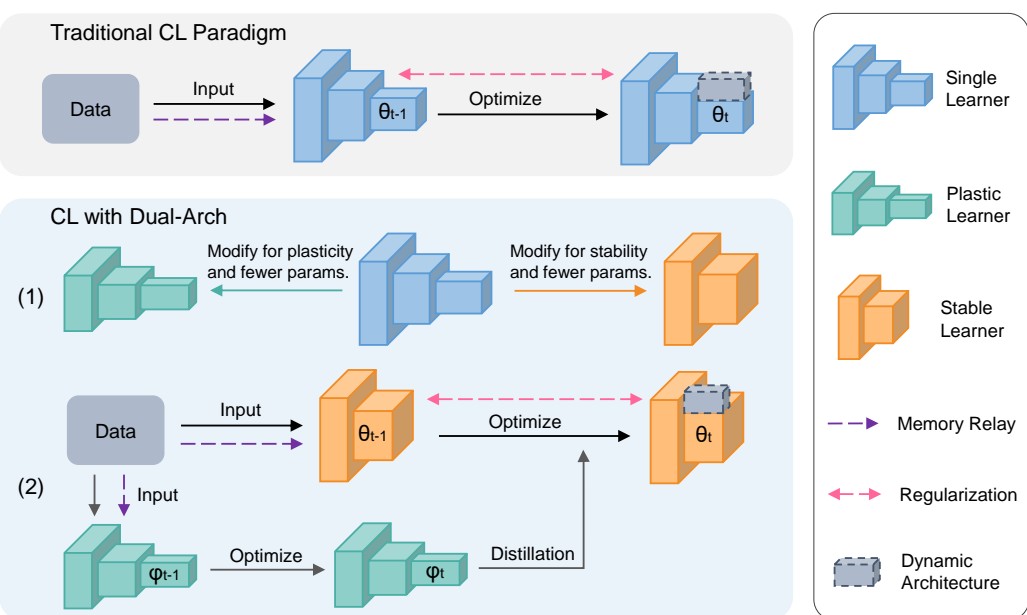

Figure 2: The formulation of the traditional CL paradigm and CL with Dual-Arch (ours). Dual-Arch (1) employs two independent learners that are designed by modifying the traditional single learner, and (2) utilizes the stable learner to perform CL with the assistance of the plastic learner. Note that Dual-Arch can be effortlessly combined with existing CL methods (denoted by the dotted lines).

(33.16% vs. 35.76%), indicating enhanced stability but diminished plasticity. Similarly, modifying the penultimate layer to increase pre-classification width yields consistent results, with AAN decreasing by 1.77% (84.64% vs. 86.41%) and FAF by 1.59% (34.17% vs. 35.76%). These observations suggest that, within a fixed parameter budget, increasing network width may enhance stability at the expense of plasticity. Conversely, the two deeper yet narrower variants exhibit slight increases in AAN, rising by 0.27% and 0.43% (86.68% and 86.87% vs. 86.41%), as well as in FAF, with increases of 0.26% and 0.22% (36.02% and 35.98% vs. 35.76%), reflecting a slight trade-off favoring plasticity over stability. This suggests that depth has a greater influence than width in plasticity under a given parameter constraint. Overall, the results reveal an inherent trade-off between stability and plasticity at the architectural level, governed by architectural design choices within specific parameter limits.

## 4 DUAL-ARCHITECTURE FRAMEWORK FOR CONTINUAL LEARNING

In this section, we propose Dual-Arch, a framework that can be easily plugged in existing CL methods, to address the stability-plasticity dilemma at the architectural level. Specifically, we will provide an overview of the Dual-Arch framework and detail its learning algorithm.

## 4.1 THE FORMULATION OF DUAL-ARCH

The overall framework of Dual-Arch is illustrated in Fig. 2. Unlike the existing CL paradigm which relies on a single learner, the Dual-Arch framework distributes the roles of plasticity and stability across two distinct models: the plastic learner and the stable learner. Inspired by existing research Kim et al. (2023) that employs auxiliary models to enhance plasticity, our framework designates the stable learner as the main model, with the plasticity learner serving as an auxiliary model. Throughout the learning process, the plastic learner is dedicated to the extraction of new knowledge, allowing for the potential forgetting of previous knowledge. Conversely, the stable learner is responsible for retaining existing knowledge while integrating new knowledge with the assistance of the plastic learner.

Dual-Arch allows the combination of the strengths of both stable and plastic architectures by employing corresponding architectures for the two learners. Specifically, these architectures are designed through targeted modifications to the original one, with the objective of enhancing plasticity or stability. Additionally, to overcome the increased memory consumption associated with incorporating an additional model, we concurrently reduce the parameter counts for both learners.

It is also worth highlighting that the Dual-Arch framework is designed to facilitate integration with a variety of CL methods, serving as a plug-and-play component. This integration can be easily achieved by applying these CL methods when training the stable learner, mirroring the training process of the single learner within the traditional CL paradigm. Furthermore, for replay-based methods, the replay buffer is concatenated with the training data for both the stable and plastic learners.

## 4.2 ARCHITECTURES FOR THE STABLE AND PLASTIC LEARNERS

This subsection presents the specific architectural designs tailored to the stable and plastic learners, with the objective of achieving superior CL performance while minimizing parameter counts. Building upon the insights from Sec. 3, we employ a wide and shallow architecture for the stable learner, denoted as Sta-Net, and a deep and thin architecture for the plastic learner, denoted as Pla-Net. Following standard practices Masana et al. (2022); Goswami et al. (2024), we have chosen ResNet-18 as the foundation for crafting both architectures. Specifically, Sta-Net retains the same width as ResNet-18 but incorporates only half as many residual blocks. Furthermore, we modify the GAP layer of Sta-Net to produce an output feature map of size $2 \times 2$ instead of the original $1 \times 1$, thereby increasing the width of the classifier. To design Pla-Net, we maintain the depth of ResNet-18 while reducing its width from 64 to 42 to align with the parameter count of Sta-Net.

## 4.3 LEARNING ALGORITHM OF DUAL-ARCH

**Preliminaries.** Before further description, some definitions related to the CL are presented. CL aims to learn from a dynamic data stream. Following convention Zhou et al. (2023a), we consider a sequence of $K$ tasks (also known as steps) $\{\mathcal{D}_1, \mathcal{D}_2, \ldots, \mathcal{D}_K\}$ without overlapping classes. Specifically, $\mathcal{D}_k \sim \{\mathcal{X}_k, \mathcal{Y}_k\}$ represents the data of the $k$-th step, containing $N_k$ classes. Here, $\mathcal{X}_k$ denotes the set of samples, and $\mathcal{Y}_k$ denotes their respective labels. At the $k$-th step, the CL model is trained on $\mathcal{D}_k^{train}$ and then tested on $\mathcal{D}_{0:k}^{test}$, which denotes the joint test dataset from task 0 to task $k$. For replay-based methods, parts of data from previous tasks are preserved and incorporated into the $\mathcal{D}_k^{train}$. In the traditional CL paradigm using a single learner, the training loss at the $k$-th step can be formulated as:

$$L_{single} = L_{CE} + L_{CL}, \tag{2}$$

where the loss term $L_{CE}$ is the classification loss calculated using a cross-entropy loss function, and $L_{CL}$ is specifically defined by the particular used CL methods. Specifically, we consider the CL learner parameterized by weights $\theta_k$ and we use $o(x)$ to indicate the output logits of the learner on input $x$. the $L_{CE}$ is defined as:

$$L_{CE}(x, y; \theta_k) = -\log \frac{\exp(o_y)}{\sum_{m=1}^{N^k} \exp(o_m)}. \tag{3}$$

The learning process of the Dual-Arch framework involves training the plastic and stable learners in sequence. In the initial stage of the learning process, our framework trains the plastic learner as a new task emerges. At this stage, the primary objective is to facilitate the acquisition of new knowledge,

without consideration of the maintenance of previously acquired knowledge. Consequently, the training objective of the plastic learner is simplified to minimize the classification loss on the current training data, i.e., $L_{plastic} = L_{CE}$. Subsequently, the stable learner is trained to integrate the existing knowledge with that acquired by the plastic learner. This process entails the transfer of recently acquired knowledge from the plastic learner to the stable learner via knowledge distillation. Specifically, a distillation loss term is incorporated into the training objective of the stable learner, to align the logit outputs between the stable and plastic learners. Following convention Hinton et al. (2015), a hard label loss (i.e., cross-entropy loss) is also employed to minimize the discrepancy between the predictions of the stable learner and the actual labels of the training data. Moreover, established CL methods are implemented during this phase to facilitate the retention of previous knowledge, which can also be expressed as a loss term. In light of the aforementioned considerations, the total learning target of the stable learner can be formulated as:

$$L_{stable} = \alpha L_{CE} + (1 - \alpha)L_{KD} + L_{CL}, \qquad (4)$$

where $L_{KD}$ denotes the distillation loss and $\alpha$ is a hyper-parameter that balances the weight of $L_{KD}$ and $L_{CE}$. We set the default value of $\alpha$ to 0.5, following Hinton et al. (2015).

Within Dual-Arch, the distillation loss $L_{KD}$ is employed to enhance the plasticity of the stable learner, which involves enabling it to learn from the soft outputs of the plastic learner. Specifically, the $L_{KD}$ is calculated by measuring the Kullback-Leibler divergence between the soft outputs of the teacher model (i.e., the plastic learner) and those of the student model (the stable learner) on the current data. Let $T$ denote the teacher model, and $S$ denote the student model, the $L_{KD}$ is defined as:

$$L_{KD} = -\sum_{i=1}^{N^k} P_T^i \log P_S^i, \qquad (5)$$

where $P_T$ and $P_S$ represent the soft outputs of the teacher and student models. These soft outputs are derived by applying the SoftMax function to transform the output logits of these models, i.e., $O_T$ and $O_S$, into probability distributions. Specifically, $P_T = \text{SoftMax}(O_T/t)$ and $P_S = \text{SoftMax}(O_S/t)$, where $t$ is the temperature factor that controls the smoothness of the soft outputs.

The detailed training procedure of the proposed framework is summarized in Alg. 1. Throughout a sequence of $N$ tasks, the training alternates between the plastic and stable learners. For each task $t$, the plastic learner is first trained using classification loss $L_{CE}$ (Lines 2-3). Once the optimal weights are obtained, this model is preserved as a teacher model for the subsequent phase (Line 4). Following this, the stable learner is trained using the loss function described in Eq. (4), as shown in Lines 5-6.

---

**Algorithm 1:** Training Procedure of Dual-Arch

**Input:** Weights of the stable learner $\theta_0$, Weights of the plastic learner $\phi_0$, Hyperparameters $\alpha$
**Output:** Optimal weights of the stable learner $\theta_K$

1 **for** *task k = 1, 2, .., K* **do**
      // Train the plastic learner
2     **for** *epoch e = 1, 2, .., E* **do**
3         Train $\phi_{k-1}$ with classification loss $L_{CE}$ on task $k$ to obtain $\phi_k$
4     Freeze and save $\phi_k$
      // Train the stable learner
5     **for** *epoch e = 1, 2, .., E* **do**
6         Train $\theta_{k-1}$ with $L_{stable}$ (Eq. (4)) on task $k$ to obtain $\theta_k$

---

## 5 EXPERIMENT

### 5.1 EXPERIMENT SETUP

**Benchmark.** Following convention Rebuffi et al. (2017), We choose CIFAR100 Krizhevsky et al. (2009) and ImageNet100 Deng et al. (2009) for evaluation. Both datasets are divided into **10** tasks of 10 classes each and **20** tasks of 5 classes each to construct four benchmarks: CIFAR100/10, CIFAR100/20, ImageNet100/10, and ImageNet100/20.

**Baselines.** To assess the efficacy of our proposed method, we integrate it into five distinct CL approaches spanning the three major categories: replay-based, regularization-based, and architecture-based methods. These methods include iCaRL Rebuffi et al. (2017), WA Zhao et al. (2020), DER Yan et al. (2021), Foster Wang et al. (2022), and MEMO Zhou et al. (2023b). Specifically, we compare the performance of Dual-Arch with that of the original ResNet-18 to evaluate the enhancements provided by our method. We also select ArchCraft Lu et al. (2024) as a baseline, which employs a single CL-friendly architecture to improve CL performance, to show the benefits of dual architectures.

**Implementation Setup.** For all experiments, we train all models by 200 epochs in the first task and 100 epochs in the subsequent tasks. The learning rate starts from 0.1 and gradually decays with a cosine annealing scheduler. A fixed memory size of 2,000 exemplars is utilized for all replay-based methods during the learning process. Given the significant impact of hyper-parameters on CL Cha & Cho (2024); Mirzadeh et al. (2020), the hyperparameters for all methods adhere to the settings in the open-source library PyCIL Zhou et al. (2023a) to ensure a fair comparison. Following convention Mirzadeh et al. (2022b); Zhou et al. (2023a), the first convolution layer and following max pooling layer of networks are replaced by a $3 \times 3$ convolution layer with a stride of 1 for CIFAR100.

**Evaluation Metrics.** The overall performance of CL is measured by two metrics: the *Last Accuracy* (LA) and the *Average Incremental Accuracy* (AIA). The LA is the total classification accuracy after the last task, which reflects the overall accuracy among all classes. Further, the AIA denotes the average classification accuracy over all tasks, which reflects the performance across all incremental steps. The higher LA and AIA, the better overall CL performance. Let $K$ be the number of tasks, these two metrics are defined as $LA = A_K$, $AIA = \frac{1}{K} \sum_{b=1}^{K} A_b$, where $A_b$ represents classification accuracy measured on the test set that covers all tasks learned up to and including the $b$-th task.

## 5.2 OVERALL RESULTS

Tab. 2 presents the comparative performance of Dual-Arch using five state-of-the-art CL methods. The results demonstrate that across various methods, datasets, and incremental steps, the integration of Dual-Arch consistently outperforms the baseline that employs ResNet-18 as a single learner. In particular, adopting Dual-Arch leads to maximum improvements of 10.29% in LA and 7.62% in AIA, while simultaneously reducing the parameter counts by at least 33%. Moreover, Dual-Arch significantly outperforms Arch-Craft in most cases, underscoring the advantages of dual architectures over a single, CL-friendly architecture. In conclusion, Dual-Arch emerges as a valuable complement to existing CL methods, enhancing both effectiveness and efficiency.

## 5.3 ABLATION STUDY

In this subsection, we present the results of our ablation study to show the significance of employing dual networks in conjunction with dedicated architectures. To simplify, we select the CIFAR-100/10 as a representative dataset and utilize AIA as the primary performance metric for our analysis.

As displayed in Tab. 3, we examine the effects of removing two pivotal components from our method. In particular, we present the outcomes of employing only a Sta-Net to underscore the necessity of the dual-networks framework. Furthermore, we present the results obtained when employing Sta-Net or Pla-Net for both learners to highlight the importance of specialized architectures. We observe from Tab. 3 that the absence of a plastic learner leads to a decrease in AIA by an average of 2.63%. Similarly, employing non-specialized architectures for two learners within Dual-Arch results in lower performance, with the AIA declining by an average of 1.74%, 0.65%, and 1.68%. These results clearly demonstrate the benefits of each component in our proposed solution.

## 5.4 PARAMETER EFFICIENCY ANALYSIS

In this subsection, we focus on evaluating the parameter efficiency of our proposed Dual-Arch by selecting two representative CL methods: DER and Foster. To assess this more comprehensively, we vary the parameter counts of Dual-Arch and ResNet-18 by reducing the network width by a quarter and a half. As illustrated in Fig. 3, the Dual-Arch series significantly outperforms the baseline in terms of parameter efficiency. Specifically, Dual-Arch can enhance AIA by 0.90% and 1.94% when using DER and Foster as the CL method, respectively, while simultaneously reducing parameter counts by 87% and 81%. Additionally, Dual-Arch surpasses ArchCraft, a state-of-the-art solution that

Table 2: The LA, AIA and FAF (%) using five state-of-the-art CL methods. '#P' represents the parameter counts of all used networks (including auxiliary networks). 'Improvement' represents the boost of Dual-Arch towards original methods. Note that the parameter counts of DER and MEMO vary from incremental settings, resulting in two values for '/20' and '/10'. **Bolded** indicates best.

| Method | #P (M) | CIFAR100/20 | | | CIFAR100/10 | | | ImageNet100/20 | | | ImageNet100/10 | | |
|---|---|---|---|---|---|---|---|---|---|---|---|---|---|
| | | LA↑ | AIA↑ | FAF↓ | LA↑ | AIA↑ | FAF↓ | LA↑ | AIA↑ | FAF↓ | LA↑ | AIA↑ | FAF↓ |
| iCaRL | 22.4 | 49.78 | 65.63 | 33.33 | 54.87 | 68.30 | 27.76 | 46.22 | 63.89 | 41.05 | 51.74 | 68.47 | 35.91 |
| w/ ArchCraft | 17.4 | **52.60** | **68.71** | - | 55.52 | 69.62 | - | 45.12 | 63.98 | - | 52.46 | 68.42 | - |
| w/ Ours | 15.1 | 52.53 | 67.80 | 29.49 | **57.69** | **70.40** | 23.63 | **47.22** | **65.06** | 35.66 | **54.84** | **69.37** | 28.22 |
| Improvement | ↓33% | +2.75 | +2.17 | -3.84 | +2.82 | +2.10 | -4.13 | +1.00 | +1.17 | -5.39 | +3.10 | +0.90 | -7.69 |
| WA | 22.4 | 46.78 | 62.75 | 19.05 | 56.98 | 69.16 | 23.53 | 46.98 | 65.76 | 39.05 | 57.64 | 71.20 | 28.27 |
| w/ ArchCraft | 17.4 | 53.23 | **69.19** | - | **59.79** | 71.40 | - | 49.94 | 67.20 | - | 58.86 | 71.56 | - |
| w/ Ours | 15.1 | **55.02** | 68.84 | 24.91 | 59.78 | **71.57** | 17.91 | **52.84** | **68.79** | 31.73 | **60.84** | **72.57** | 24.53 |
| Improvement | ↓33% | +8.24 | +6.09 | +5.86 | +2.80 | +2.41 | -5.62 | +5.86 | +3.03 | -7.32 | +3.20 | +1.37 | -3.74 |
| DER | 224.4/112.2 | 58.39 | 70.19 | 25.63 | 61.83 | 72.48 | 22.13 | 64.32 | 74.91 | 20.51 | 67.40 | 75.93 | 15.22 |
| w/ ArchCraft | 173.5/86.8 | 61.65 | 73.59 | - | 63.94 | 74.84 | - | 63.98 | 74.50 | - | 68.34 | 77.26 | - |
| w/ Ours | 106.9/55.9 | **64.08** | **73.86** | 20.08 | **66.22** | **75.08** | 17.73 | **65.40** | **75.17** | 16.97 | **68.52** | **77.49** | 12.96 |
| Improvement | ↓52%/50% | +5.69 | +3.67 | -5.55 | +4.39 | +2.60 | -4.40 | +1.08 | +0.26 | -3.54 | +1.12 | +1.56 | -2.26 |
| Foster | 22.5 | 49.99 | 63.39 | 35.03 | 58.67 | 69.95 | 27.39 | 54.74 | 66.77 | 34.67 | 62.88 | 71.09 | 25.18 |
| w/ ArchCraft | 17.5 | 57.22 | 69.99 | - | **61.44** | 72.54 | - | 54.32 | 66.41 | - | 61.94 | 71.16 | - |
| w/ Ours | 15.4 | **57.69** | **71.01** | 23.75 | 61.23 | **73.22** | 18.23 | **55.20** | **67.63** | 32.42 | **63.24** | **72.42** | 25.04 |
| Improvement | ↓32% | +7.70 | +7.62 | -11.28 | +2.56 | +3.27 | -9.16 | +0.46 | +0.86 | -2.25 | +0.36 | +1.33 | -0.14 |
| MEMO | 171.7/87.2 | 52.10 | 67.60 | 35.71 | 58.46 | 70.71 | 27.99 | 56.10 | 69.13 | 29.64 | 61.64 | 73.31 | 21.87 |
| w/ ArchCraft | 126.6/64.6 | 57.28 | 72.07 | - | 61.93 | 73.30 | - | 57.46 | 70.54 | - | 62.46 | 74.01 | - |
| w/ Ours | 101.1/53.1 | **62.39** | **72.69** | 24.09 | **65.35** | **74.34** | 20.82 | **60.26** | **72.53** | 24.59 | **65.40** | **75.54** | 16.53 |
| Improvement | ↓41%/39% | +10.29 | +5.09 | -11.62 | +6.89 | +3.63 | -7.17 | +4.16 | +3.40 | -5.05 | +3.76 | +2.23 | -5.34 |

Table 3: The ablation study results using five CL methods. We report the mean±std of 3 runs with different initializations. † denotes performing CL with a single learner. **Bolded** indicates the best.

| Stable Learner | Plastic Learner | AIA (%) on CIFAR100/10 | | | | | |
|---|---|---|---|---|---|---|---|
| | | iCaRL | WA | DER | Foster | MEMO | **Average** |
| Sta-Net | Pla-Net | **70.21**±0.19 | **71.53**±0.13 | **75.26**±0.20 | **73.18**±0.04 | **74.44**±0.07 | **72.92** |
| Sta-Net | None† | 66.69±0.10 | 69.33±0.17 | 72.47±0.07 | 70.84±0.24 | 72.11±0.27 | 70.29 (-2.63) |
| Pla-Net | Pla-Net | 69.57±0.10 | 69.98±0.08 | 74.64±0.28 | 71.92±0.28 | 69.80±0.15 | 71.18 (-1.74) |
| Sta-Net | Sta-Net | 69.27±0.26 | 71.38±0.25 | 74.30±0.08 | 72.65±0.11 | 73.77±0.05 | 72.27 (-0.65) |
| Pla-Net | Sta-Net | 69.85±0.13 | 70.31±0.26 | 74.25±0.35 | 71.86±0.06 | 69.92±0.12 | 71.24 (-1.68) |

enhances parameter efficiency in CL by recrafting the network architecture. These empirical results highlight the potential of Dual-Arch to significantly benefit CL in memory-restricted scenarios.

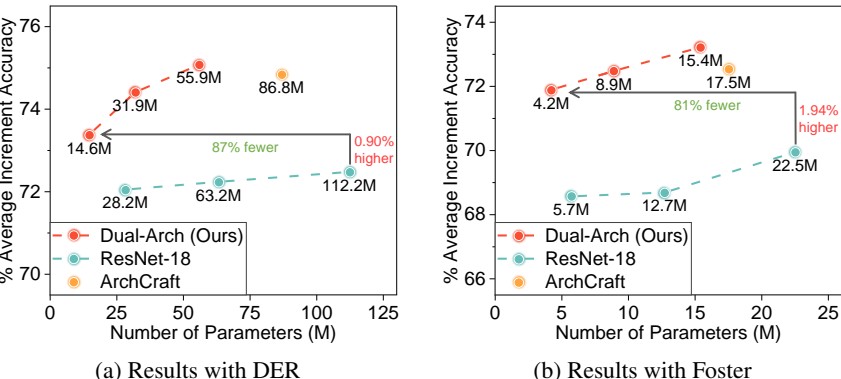

(a) Results with DER

(b) Results with Foster

Figure 3: **Performance of CL vs. Number of Parameters** using DER and Foster on CIFAR100/10.

## 5.5 ANALYSIS ON THE STABILITY-PLASTICITY TRADE-OFF

To further scrutinize the effectiveness of Dual-Arch in combining the strength of both architectures, we compare it to a single learner with one of the architectures (i.e., Pla-Net or Sta-Net). To simplify,

we choose the top-performing approach DER as the used CL method. We observe from Fig. 4 (a) that Dual-Arch achieves the best overall performance in CL. Moreover, as illustrated in Fig. 4 (b) and (c), the single learner either forgets severely on previous tasks (Pla-Net) or underperforms on new ones (Sta-Net), whereas Dual-Arch demonstrates competitive performance in both aspects. This result indicates that Dual-Arch combines the advantages of both types of architecture, leading to a trade-off between stability and plasticity at the architectural level.

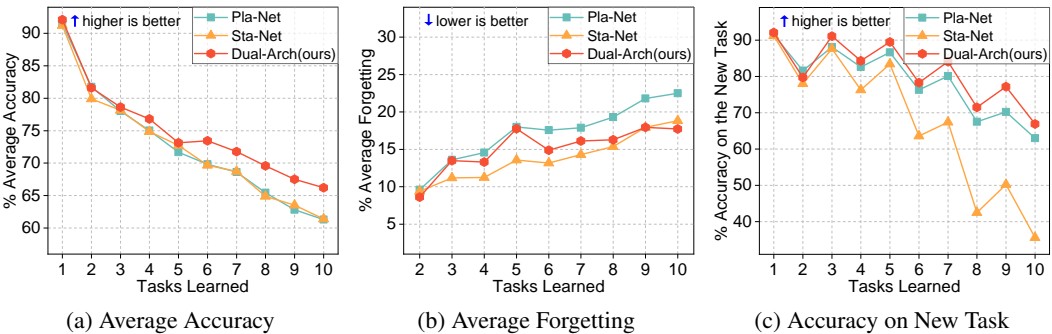

(a) Average Accuracy  (b) Average Forgetting  (c) Accuracy on New Task

Figure 4: The performance of Dual-Arch and two baselines using DER on CIFAR100/10.

## 5.6 ANALYSIS ON BIAS-CORRECTION

In *Class IL*, the task-recency bias is a major cause of catastrophic forgetting, where models tend to misclassify instances from earlier tasks as belonging to more recently introduced classes during inference Masana et al. (2022); Zhao et al. (2020). To discover the reasons why Dual-Arch benefits CL, we further evaluate its effectiveness in mitigating the task-recency bias. Specifically, we present the task confusion matrices for the Dual-Arch and the baseline which employs a single ResNet-18 in Fig. 5. From Fig. 5 (b) and (d), we observe that the integration of Dual-Arch facilitates a more precise determination of the correct task ID, thereby reducing inter-task classification errors. Notably, Dual-Arch significantly diminishes the misclassification of data from earlier tasks (e.g., task 1) as belonging to recently learned tasks (e.g., task 10). These observations indicate that Dual-Arch can effectively reduce the task-recency bias, thus mitigating catastrophic forgetting.

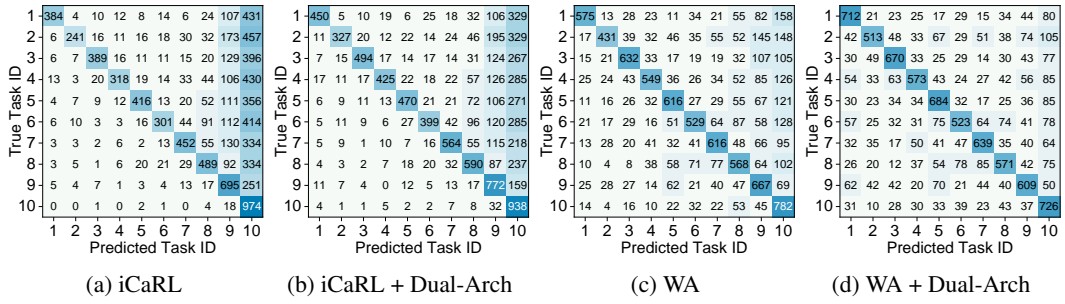

(a) iCaRL  (b) iCaRL + Dual-Arch  (c) WA  (d) WA + Dual-Arch

Figure 5: Task confusion matrices after learning the final task of different CL methods w/ and w/o Dual-Arch plugged in on CIFAR100/10. Results of other methods are reported in Appendix A.2.

## 6 CONCLUSION

In this paper, we point out the stability-plasticity dilemma at the architectural level and further introduce Dual-Arch, a novel CL framework, to address it. Dual-Arch is built on a parallel level with most of the existing CL methods (architecture vs. parameter), thereby serving as a plug-in component for enhancing CL. Our extensive experiments demonstrate that Dual-Arch consistently outperforms the baselines while significantly reducing the parameter counts. We hope this work inspires further study on exploring a better trade-off between stability and plasticity from an architectural perspective.

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

# A APPENDIX

## A.1 COMPARISON ON PARAMETER COUNTS AND FLOPS

We compute the sum of the parameter counts of all used models for each incremental step and report their peak values throughout the CL process as the final result. For instance, for iCaRL, we sum the parameter counts of the current and last models. For iCaRL with Dual-Arch, we sum the parameter counts of the plastic learner, the current stable learner, and the last stable learner. All results are detailed in Tab. 2 of the main paper. Moreover, it should be noted that the parameter counts vary slightly between the CIFAR100 and ImageNet100, and we report all results based on CIFAR100.

Additionally, we note that despite Dual-Arch involving the training of two models, the total FLOPs remain less than those of the baselines. Taking CIFAR100 as an example, the FLOPs for Sta-Net and Pla-Net are 255M and 241M, respectively, resulting in a combined total of 496M. In contrast, the FLOPs for ResNet-18 and ResAC are approximately 558M and 1383M, respectively.

## A.2 ADDITIONAL RESULTS ON BIAS-CORRECTION

We report the task confusion matrices for the DER, Foster and MEMO with/without Dual-Arch here.

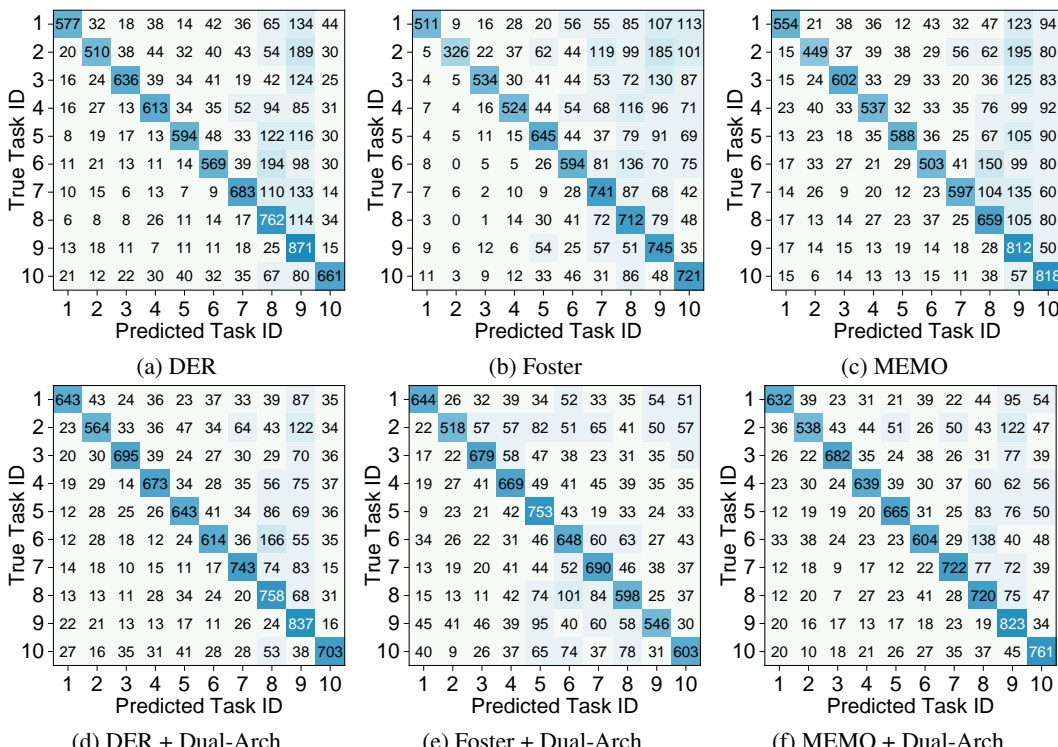

Figure 6: Task confusion matrices after learning the final task of different CL methods w/ and w/o Dual-Arch plugged in on CIFAR100/10.

### A.3 IMPLEMENTATION DETAILS

We employ the same data augmentation as in PyCIL Zhou et al. (2023a) for all experiments. For the experiments in Section 3, we report results in different task orders using 5 seeds 1, 2, 3, 4, and 5. For other experiments, we adhere to a fixed seed of 1993, consistent with established conventions Rebuffi et al. (2017); Zhou et al. (2023a). We utilize a temperature factor for Dual-Arch of $t = 4$ for CIFAR100 and $t = 3$ for ImageNet-100.

### A.4 ARCHITECTURAL DIMENSIONS OF STABILITY AND PLASTICITY IN MLP

We further investigate the impact of network width (i.e., the number of neurons in the hidden layer) and depth (i.e., the number of layers) on the stability and plasticity of the MultiLayer Perceptron (MLP). Following HAT Serra et al. (2018), we employ a width of 800 and a depth of 4 as the default design for the MLP. Additionally, we design a wider yet shallower variant and a deeper yet thinner variant, both with a parameter count comparable to the default design. We evaluated all MLPs on the split MNIST dataset, which consists of five tasks, using the LWF Li & Hoiem (2017). Note that we train all models with 10 epochs, and report results using five different task orders. The results are reported in Tab. 4. We observe that the wider yet shallower variant exhibits lower values for both AAN and FAF. These results suggest that within a fixed parameter budget, the wider and shallower variants offer superior stability at the expense of reduced plasticity, a trend consistent with the findings observed in ResNet architectures.

Table 4: The AAN and FAF (%) of MLP with different depths and widths. Note that the '#P' denotes the parameter counts of a single architecture here.

| Depth | Width | #P | AAN ↑ | FAF ↓ |
|-------|-------|------|---------------------|------------------------|
| 4 | 800 | 1.92 | 84.20±5.37 | 42.10±7.58 |
| 3 | 1050 | 1.94 | 79.23±5.13 (-4.97) | 26.78±5.18 (-15.32) |
| 5 | 680 | 1.93 | 87.35±3.77 (+3.15) | 59.28±6.71 (+17.18) |

### A.5 VALIDATION ON CIFAR100/50

In this subsection, we report the results on settings with a greater number of tasks, specifically CIFAR100/50, which contains 50 tasks, in Tab. 5. These results demonstrate that Dual-Arch consistently outperforms the baselines in this challenging setting, thereby underscoring its generality.

Table 5: The LA and AIA (%) using five state-of-the-art CL methods on CIFAR100/50. **Bolded** indicates the best.

| Method | iCaRL | | WA | | DER | | Foster | | MEMO | |
|--------|-------|-------|-------|-------|-------|-------|-------|-------|-------|-------|
| | LA | AIA | LA | AIA | LA | AIA | LA | AIA | LA | AIA |
| Original | 45.30 | 63.99 | 42.12 | 58.26 | 55.73 | 69.53 | 43.45 | 59.81 | 42.44 | 62.57 |
| w/ ArchCraft | 48.70 | **67.24** | 39.83 | 61.02 | 57.89 | 71.53 | 53.02 | **68.14** | 54.47 | 69.96 |
| w/ ours | **48.95** | 65.93 | **47.13** | **64.41** | **61.88** | **73.09** | **53.16** | 67.83 | **58.09** | **71.17** |

### A.6 VALIDATION ON CL WITH BLURRY TASKS BOUNDARIES

Beyond Class-IL, a series of works have focused on a more challenging and realistic CL scenario where task boundaries are not explicitly available, known as Generalized Class IL Buzzega et al. (2020); Arani et al. (2022). In this section, we validate the generality of Dual-Arch in this setting. Following convention Arani et al. (2022); Sarfraz et al. (2022), we report the results on the typical benchmark, GCIL-CIFAR-100, as shown in Tab. 6. Our findings indicate that Dual-Arch consistently enhances CL performance in this scenario, underscoring its broad applicability.

### A.7 VALIDATION ON VISION TRANSFORMERS

While our study primarily focuses on ResNet, the insights presented in our paper are potentially applicable to other architectures, such as Vision Transformers (ViTs). In this subsection, we report

Table 6: The LA (%) on GCIL-CIFAR-100 with different buffer sizes. **Bolded** indicates the best. Note that the benchmark settings follow Arani et al. (2022).

| Method | Buffer Size 500 | Buffer Size 1000 |
|---|---|---|
| ER Rostami et al. (2019) | 20.30 | 34.13 |
| w/ Dual-Arch (ours) | **27.57** (+7.27) | **35.40** (+1.27) |
| DER++ Buzzega et al. (2020) | 25.82 | 33.64 |
| w/ Dual-Arch (ours) | **30.34** (+4.52) | **36.84** (+3.20) |

the results of transferring our method to SepViT Li et al. (2022) on ImageNet100/10, with all models trained from scratch. Note that the training settings are consistent with Sec. 5.1, but the learning rate and optimizer are adjusted to match the official implementation of SepVit Li et al. (2022). The results, presented in Tab. 7, demonstrate that Dual-Arch consistently enhances the CL performance of SepViT, indicating its generality to ViTs.

Table 7: The LA and AIA (%) using SepVit on ImageNet100/10. '#P' represents the parameter counts of all used networks (including auxiliary networks). **Bolded** indicates the best.

| Method | #P (M) | LA | AIA |
|---|---|---|---|
| iCaRL | 7.57 | 43.08 | 60.62 |
| w/ ours | 5.32 | **46.34** (+3.26) | **63.09** (+2.47) |
| WA | 7.57 | 38.40 | 57.67 |
| w/ ours | 5.32 | **44.28** (+5.88) | **61.15** (+3.48) |

## A.8 ARCHITECTURAL DIMENSIONS OF STABILITY AND PLASTICITY IN VISION TRANSFORMERS

We further investigate the impact of network width (i.e., the dimension of the attention heads) and depth (i.e., the number of blocks) on the stability and plasticity of ViTs. Specifically, we use SepViT-Lite Li et al. (2022) as the default design, which is configured with a width of 32 and a depth of 11. Additionally, we design a wider yet shallower variant with a width of 49 and depth of 5, which has a parameter count comparable to the default design. Both ViTs are evaluated on ImageNet-100/10 using iCaRL as the learning method Rebuffi et al. (2017). Note that the training settings are consistent with Sec. 5.1, but the learning rate and optimizer are adjusted to match the official implementation of SepVit Li et al. (2022). The results are reported in Tab. 8. We observe that the wider yet shallower variant exhibits lower values for both AAN and FAF. These results suggest that within a fixed parameter budget, the wider and shallower variants offer superior stability at the expense of reduced plasticity, a trend consistent with the findings observed in ResNet architectures.

Table 8: The AAN and FAF (%) of SepVit with different depths and widths. Note that the '#P' denotes the parameter counts of a single architecture here.

| Depth | Width | #P | AAN ↑ | FAF ↓ |
|---|---|---|---|---|
| 11 | 32 | 3.78 | 79.54 | 40.51 |
| 5 | 49 | 3.76 | 78.96 (-0.58) | 39.47 (-1.04) |

