# OpenReview forum: "Rethinking the Stability-Plasticity Trade-off in Continual Learning from an Architectural Perspective"
_ICLR.cc/2025/Conference — Submitted to ICLR 2025_

### Official Review · Reviewer_a2rA · 2024-11-01

**Soundness:** 3
**Presentation:** 4
**Contribution:** 3
**Rating:** 6
**Confidence:** 5

**Summary:**

This paper studies the stability-plasticity dilemma in continual learning from an architectural perspective. Through empirical studies, the authors demonstrate that network architecture impacts learning dynamics: deeper networks exhibit better plasticity while wider networks show better stability. Based on this insight, they propose Dual-Arch, a method using two specialized networks with complementary strengths: a deeper, thinner network optimized for plasticity and a wider, shallower network designed for stability. The framework operates by having the plastic network learn new tasks with flexibility, then transferring this knowledge to the stable network through knowledge distillation. A key feature of Dual-Arch is its ability to serve as a plug-in component for existing continual learning methods. Through extensive experiments on class incremental learning setting, the authors demonstrate that their approach not only improves performance but also achieves significant parameter efficiency compared to baseline methods, while addressing the task-recency bias.

**Strengths:**

-  I found the method simple yet pretty intuitive and results look promising as well.
- Related work is comprehensive, explaining past work on forgetting and architecture (e.g., depth, width, and network components). They also cover complementary learning system-inspired works, which is highly related.
- The paper is well-written with a clear structure
- Experiments are detailed with thorough analysis of the algorithm

**Weaknesses:**

- While parameter efficiency is emphasized, the approach is resource-intensive in terms of compute. The stable network requires waiting for the plastic network to complete learning, which, intuitively, should double the runtime compared to standard replay methods. Reporting runtime or FLOP counts would be helpful. Re-running the experiments and recording clock time may not be feasible during the rebuttal period. However, I would suggest the authors add at least FLOP counts and discuss whether their algorithm may end up taking more time.
- The analysis using iCaRL in Section 3 is somewhat counterintuitive. Although iCaRL is a well-established approach, it has distinctive features like knowledge distillation and prototype-based classification, which may obscure the architecture's direct impact on stability and plasticity. I would expect using a simpler, more standard algorithm, such as experience replay, to provide a clearer understanding of these effects. I don't suggest the authors try to repeat the experiments in a short period of time. However, it would be good to discuss why iCaRL was chosen beyond saying it is a classical replay method.
- Section 3’s results largely align with existing findings, such as those presented in "Wide Neural Networks Forget Less Catastrophically." So, I do not think they add much to existing knowledge.

**Questions:**

- Dual-Arch appears to heavily rely on ("exploit") the class incremental setup.  For instance, how does the model perform predictions within a task? And what if task boundaries are not available? Although many methods in the literature rely on this kind of information, I believe a truly continual learning model should ideally be ready to make predictions at any stage and be more robust. How could authors address these limitations?
- In relation to the "plastic learner," did the authors consider resetting its weights periodically? Previous work, such as https://arxiv.org/abs/1903.04476 and https://arxiv.org/abs/2206.09117 suggests that resetting weights can improve model plasticity.
- Vision transformers are increasingly being used in continual learning (e.g., L2P, DualPrompt, CODA-Prompt). While these methods typically rely on frozen pretrained weights rather than directly training the architecture, I'm curious about how the insights regarding depth and width presented in this paper could transfer to vision transformer architectures. What would be the architectural implications for transformers in terms of stability-plasticity trade-off?
- Comment: Section 5.3 has a typo in "ABALATION STUDY."

---

> ### Author Response · Authors · 2024-11-23
> **Initial Response**
>
> We appreciate the reviewer's recognition of the simple yet effective approach, promising performance results,  detailed experiments and analysis, and the clarity of our paper. To further address the reviewer's concern, we have:
>
> - included an analysis of FLOPs (for Weakness 1).
> - provided an explanation of why iCaRL is chosen in Section 3 (for Weakness 2).
> - provided further clarification to highlight the unique value of Section 3 (for Weakness 3).
> - included experiments on a more realistic scenario and ViTs (for Question 1,3).
> - provided an analysis of the impact of resetting the plastic learner (for Question 2).
> - proofread the paper carefully to correct typos (for Question 4).
>
> > Weakness1: Details about FLOPs.
>
> We appreciate the reviewer's valuable comment regarding the computational cost. We note that despite Dual-Arch involving the training of two models, the total FLOPs remain lower than those of the baselines. For instance, on CIFAR100, the FLOPs for Sta-Net and Pla-Net are 255M and 241M, respectively, resulting in a combined total of 496M. In contrast, the FLOPs for ResNet-18 and ResAC are 558M and 1383M, respectively. Actually, we have previously reported the FLOPs results for a batch (128) of samples in A.1 of the appendix of our initial submission, and we have revised these results to reflect the FLOPs per single sample, as reported above.
>
> > Weakness2: Concern about the use of iCaRL in Section 3.
>
> We appreciate the reviewer's thoughtful consideration and acknowledge Experience Replay (ER) is a more intuitive choice.  However, we argue that vanilla ER might not be suitable for evaluating the CL performance (especially stability) of architectures. Specifically, when using ER without other CL strategies,  a model's performance on old tasks might be mainly enhanced by learning from the replay buffer rather than preserving existing knowledge.
>
> To illustrate this point, we introduced an ablation baseline for both vanilla ER and iCaRL, denoted as 'w/ buffer only'. This baseline involves training the models exclusively on the last 5 tasks, with an initial replay buffer constructed from the first 5 tasks. We then evaluate their performance on the first 5 tasks after the last task is learned.  Here are the results on CIFAR100/10:
>
> | Method         | Task-1 | Task-2 | Task-3 | Task-4 | Task-5 | Avg           |
> | -------------- | ------ | ------ | ------ | ------ | ------ | ------------- |
> | vanilla ER     | 39.0   | 28.0   | 44.8   | 35.9   | 44.7   | 38.48         |
> | w/ buffer only | 37.6   | 26.3   | 40.1   | 35.2   | 42.7   | 36.38 (-2.10) |
> | iCaRL          | 51.0   | 36.7   | 54.4   | 44.8   | 56.3   | 48.64         |
> | w/ buffer only | 43.9   | 28.6   | 46.9   | 39.3   | 48.3   | 41.40 (-7.14) |
>
> The results show that the ablation baseline achieves comparable (for such a weak baseline) performance on tasks 1-5 as the original ER, even though it does not train on these tasks. In contrast, the performance difference in iCaRL is significantly larger than that in ER (7.14% vs. 2.10%). This demonstrates that iCaRL enables the model to better preserve existing knowledge beyond learning from the buffer, making it more effective for evaluating stability.
>
> We hope that this additional analysis highlights the limitations of ER in evaluating the CL performance, particularly stability, of architectures and provides a clearer understanding of why we did not choose it.
>
> > Weakness3: Concern about the unique findings in Section 3.
>
> While it is true that our results align with existing findings about the impact of architecture on stability, we believe they contribute to the field by extending the research to a more detailed understanding of the balance between stability and plasticity. Unlike previous works that primarily focus on overall CL performance or stability, our study delves into the separate aspects of stability and plasticity. This distinction allows us to highlight the trade-off between these two critical factors at the architectural level, which we believe adds a new dimension to the existing body of knowledge.

---

> ### Author Response · Authors · 2024-11-23
> **Initial Response -- part 2**
>
> >Question1: Experiments on more general continual learning settings.
>
> Thank you for your insightful comments. While the focus of this study is on class-incremental learning, we recognize the importance of more realistic scenarios. We have conducted additional experiments on general continual learning [1] [2] [3], a more challenging and realistic continual learning scenario where task boundaries are not explicitly available. These results indicate that Dual-Arch consistently enhances CL performance in this scenario, underscoring its broad applicability. We have also included these results in Section A.6 of this revision. Additionally, we have revised our Related Work to extend its scope by including a discussion of this scenario.
>
> | Method              | Buffer Size 500   | Buffer Size 1000  |
> | ------------------- | ----------------- | ----------------- |
> | ER [4]              | 20.30             | 34.13             |
> | w/ Dual-Arch (ours) | **27.57 (+7.27)** | **35.40 (+1.27)** |
> | DER++ [3]           | 25.82             | 33.64             |
> | w/ Dual-Arch (ours) | **30.34 (+4.52)** | **36.84 (+3.20)** |
>
> >Question2: Concern about whether resetting weights benefits plasticity in Dual-Arch.
>
> We appreciate the reviewer's insightful comments regarding the potential benefits of periodically resetting the weights of the plastic learner. We did explore this strategy but found that it did not enhance performance within our proposed framework. While we acknowledge the valuable insights from existing works that suggest that resetting weights can improve model plasticity in certain scenarios, we believe that preserving the existing weights allows the plastic learner to leverage prior knowledge for acquiring new knowledge more effectively.
>
> Here are the performance results (%AIA) of resetting the plastic learner before learning each task, denoted as 'w/ resetting,' on CIFAR100/10:
>
> | Method            | iCaRL     | WA        | DER       | Foster    | MEMO      |
> | ----------------- | --------- | --------- | --------- | --------- | --------- |
> | Dual-Arch  (ours) | **70.40** | **71.57** | **75.08** | **73.22** | **74.34** |
> | w/ resetting      | 68.25     | 70.70     | 72.96     | 72.30     | 72.12     |
>
> > Question3:Exploration and insights for vision transformer.
>
> While our study focuses on ResNet, we believe that the architectural insights presented in our paper could potentially extend to Vision Transformers (ViTs). Validation of fully transferring our solution to the pre-trained ViTs is beyond our current resources, but we have conducted preliminary experiments to validate the partial transfer of our insights to vision transformer architectures.
>
> Firstly, regarding the methods that use a frozen pretrained backbone, the trainable part of the architectures is solely a single layer known as prompt, which might make the architecture very stable but lacking in plasticity. Therefore, applying our solution by employing the pretrained ViT as the stable learner and a plastic architecture as the plastic learner may benefit continual learning. Here, we combine our proposed framework with these methods by using the ResNet-18 as the plastic learner, and the results are as follows.
>
> | Method              | CIFAR100/20 LA (%) | CIFAR100/20 AIA (%) | CIFAR100/10 AIA (%) | CIFAR100/10 AIA (%) |
> | ------------------- | ------------------ | ------------------- | ------------------- | ------------------- |
> | L2P [6]             | 73.60              | 81.52               | 79.98               | 85.67               |
> | w/ Dual-Arch (ours) | **75.96 (+2.36)**  | **83.45 (+1.93)**   | **80.65 (+0.67)**   | **86.32 (+0.65)**   |
> | DualPrompt [7]      | 75.47              | 82.64               | 80.77               | 87.02               |
> | w/ Dual-Arch (ours) | **75.94 (+0.47)**  | **83.30 (+0.66)**   | **82.20 (+1.43)**   | **87.54 (+0.52)**   |
>
> Moreover, we note that our proposed framework can also benefit ViTs within the setting of training from scratch. Here are the validation results on SepViT [4].
>
> | Method              | Params (M) | LA (%)            | AIA (%)           |
> | ------------------- | ---------- | ----------------- | ----------------- |
> | iCaRL               | 7.57       | 43.08             | 60.62             |
> | w/ Dual-Arch (ours) | 5.32       | **46.34 (+3.26)** | **63.09 (+2.47)** |
> | WA                  | 7.57       | 38.40             | 57.67             |
> | w/ Dual-Arch (ours) | 5.32       | **44.28 (+5.88)** | **61.15 (+3.48)** |
>
> Finally, a possible approach for fully transferring our proposed framework to pretrained ViTs might involve designing two distinct vision transformer architectures—one wide and shallow, the other deep and thin—and pre-training them on a large dataset. These pretrained models could then be utilized in Dual-Arch, similar to what we have explored in our study. However, the computational cost associated with pre-training these models is substantial and currently beyond our resources.

---

> ### Author Response · Authors · 2024-11-23
> **Initial Response -- part 3**
>
> >Question4: typo in Section 5.3
>
> We have corrected the typo, changing "ABALATION STUDY" to "ABLATION STUDY." We have thoroughly proofread the manuscript to address every other issue we found. Thank you for pointing this out.
>
> **References.**
>
> [1] Arani et al. Learning fast, learning slow: A general continual learning method based on complementary learning system.
>
> [2] Sarfraz et al. SYNERgy between SYNaptic Consolidation and Experience Replay for General Continual Learning
>
> [3] Buzzega et al. Dark experience for general continual learning: a strong, simple baseline.
>
> [4] Rostam et al.  Complementary learning for overcoming catastrophic forgetting using experience replay.
>
> [6] Wang et al. Learning to prompt for continual learning.
>
> [7] Wang et al. Dualprompt: Complementary prompting for rehearsal-free continual learning.

---

> ### Author Response · Authors · 2024-11-27
> **A Gentle Reminder of Feedbacks**
>
> Dear Reviewer a2rA:
>
> Thanks for your careful comments and your time spent on our work. We have revised our paper and added the discussion and experiments concerning.
>
> Currently, all of your concerns can be resolved in the responses and revised version of the paper. However, as the revision deadline is approaching, we kindly request your feedback to confirm that our responses and revisions have effectively resolved your concerns. If there are any remaining issues, we would be grateful for the opportunity to address them to ensure the quality of our work.
>
> Yours sincerely,
>
> Authors of Paper 6928

---

> ### Author Response · Authors · 2024-12-02
>
> Dear Reviewer a2rA,
>
> Hope you had a wonderful Thanksgiving holiday week! We would like to sincerely thank you once again for your valuable comments and your time spent on our work. We believe that all your questions can be well addressed in the responses and revised version of the paper.
>
> As the extended discussion period ends tomorrow, we would greatly appreciate your feedback soon. If our rebuttal adequately addresses your concerns, we kindly request an update on your evaluations. Once again, thank you very much for your reviews and feedback!
>
> Yours sincerely,
>
> Authors of Paper 6928

---

### Official Review · Reviewer_e7KZ · 2024-11-01

**Soundness:** 2
**Presentation:** 2
**Contribution:** 2
**Rating:** 3
**Confidence:** 4

**Summary:**

The paper investigates the trade-off between plasticity and stability in neural networks and introduces an approach that leverages two  architectures (sta-net and pla-net) to address this balance. Experimental results across multiple datasets and various continual learning  methods demonstrate the improvements achieved with this approach, highlighting its effectiveness and potential for enhancing model performance.

**Strengths:**

* The paper provides an interesting exploration of the tradeoff between plasticity and stability at an architectural level under a fixed parameter count constraint. The insight that deeper networks enhance plasticity while wider networks favor stability is particularly compelling can help in designing more balanced models in the future.
* The dual-arch model using two models one for plasticity and other for stability is simple and effective as shown in experiments on various datasets.
* The paper is clear and well-organized.

**Weaknesses:**

* **Motivation and presentation.** The paper’s motivation could be strengthened. Specifically, Table 1 is challenging to interpret, as it lacks a total parameter count despite referencing it in the text. Additionally, the table format makes it difficult to assess the isolated effects of depth and width, as both vary simultaneously across rows.
* **Details on Hyper-Parameters:** More transparency around hyper-parameter tuning is needed, particularly for $\alpha$ and $\beta$. Various works [1,2] have emphasized the impact of hyper-parameter tuning in continual learning, and it is important to discuss the assumptions and the setup in the paper. Further, could the authors provide confidence intervals for the methods? Without error bars, it’s hard to fully justify the two-architecture approach. For example, in Table 3, the performance difference is minimal (less than a percent) compared to simply using two stable network architectures.
* **Comparison with Related Work:** The paper would benefit from a more comprehensive comparison with works [3, 4, 5] that use methods like distillation or network expansion to address similar issues in continual learning with better performance. These comparisons are essential to clarify the unique contributions of this study and position it with respect to prior works.

References.
[1] Mirzadeh et al., Understanding the Role of Training Regimes in Continual Learning.
[2] Cha et al., Hyperparameters in Continual Learning: A Reality Check.
[3] Madaan et al., Heterogeneous Continual Learning.
[4] Yoon et al., Lifelong Learning with Dynamically Expandable Networks.
[5] Sun et al., Decoupling Learning and Remembering: a Bilevel Memory Framework with Knowledge Projection for Task-Incremental Learning.

**Questions:**

* For KD, do you use the current dataset?
* Can you report forgetting in Table 2 as reducing forgetting is a key focus of the work?
* Regarding the ablation studies, how are the stability (sta-net) and plasticity (pla-net) architectures trained? Specifically, how is the loss modified from Eq. (2)? Clearer details would help readers better understand Figure 4 and Table 3.

---

> ### Author Response · Authors · 2024-11-23
> **Initial Response**
>
> We appreciate the reviewer's recognition of our interesting findings, simple yet effective approach, and the clarity of our paper. To further address the reviewer's concern, we have:
>
> - revised Table 1 to better strengthen our motivation (for Weakness 1).
> - provided more detailed hyper-parameters settings and error bars for Table 3 (for Weakness 2).
> - included a more detailed comparison with related works (for Weakness 3).
> - provided a more detailed workflow of our approach (for Question 1,3)
> - included forgetting results for Table 2 (for Question 2).
>
> >Weakness1: Presentation of Table 1.
>
> Thank you for your suggestion to include the total parameter count in Table 1 to enhance clarity. The revised table is as follows.
>
> | Depth | Width | Penultimate Layer | Params (M) | AAN (%) *↑*            | FAF (%) *↓*            |
> | ----- | ----- | ----------------- | ---------- | ---------------------- | ---------------------- |
> | 18    | 64    | GAP               | 11.23      | 86.41$\pm$0.60         | 35.76$\pm$1.62         |
> | 10    | 96    | GAP               | 11.10      | 83.44$\pm$0.84 (-2.97) | 33.16$\pm$1.28 (-2.60) |
> | 18    | 64    | 4 *×* 4 AvgPool   | 11.38      | 84.64$\pm$0.43 (-1.77) | 34.17$\pm$2.03 (-1.59) |
> | 26    | 52    | GAP               | 11.56      | 86.68$\pm$0.70 (+0.27) | 36.02$\pm$1.79 (+0.26) |
> | 34    | 46    | GAP               | 11.04      | 86.87$\pm$0.54 (+0.43) | 35.98$\pm$1.97 (+0.22) |
>
> Note that the parameter counts of the variant differ only slightly (less than $\pm$3%) from the original ResNet-18.
>
> The reason we vary both width and depth simultaneously is to maintain a consistent number of parameters. This ensures that any observed differences in performance are mainly attributable to architectural designs rather than parameter counts. Changing the width or depth individually would significantly alter the parameter counts, introducing an additional variable. Our study's motivation stems from the observation that, with similar numbers of parameters, wider and shallower networks tend to exhibit greater stability but lower plasticity, while deeper and narrower networks show the opposite trend.  The experiments in Table 1 effectively demonstrate this trade-off.

---

> ### Author Response · Authors · 2024-11-23
> **Initial Response -- part 2**
>
> >Weakness2: Details on hyper-parameters.
>
> We highly recognize the significant impact of hyper-parameter tuning on Continual Learning (CL). To ensure a fair comparison and avoid over-tuning, we have adhered to the following practices:
>
> - For parameter $\alpha$, we set it to 0.5 as mentioned in Section 4.3, which aligns with the earliest work on knowledge distillation [1]. This setting is consistent across experiments that use dual architectures, including the ablation study, to ensure a fair comparison. We have further highlighted this content in Section 4.3 of this revision.
> - For parameter $\beta$, it is determined by the CL methods in combination with our work, as detailed in Section 4.3. We followed the default settings in the open-source library PyCIL [2] for both Dual-Arch and all baselines to ensure a fair comparison, as mentioned in the Implementation Setup in Section 5.1. We have further highlighted the importance of hyper-parameters in Section 5.1 of this revision.
>
> Moreover, considering that $\beta$ is actually not related to our study, we have removed it from Eq.(2), now presented as Eq. (4) in the current revision, to avoid misunderstanding. Below are the performance results (%AIA) with different values of $\alpha$.
>
> | Settings                          | iCaRL CIFAR100/20 | iCaRL CIFAR100/10 | WA CIFAR100/20 | WA CIFAR100/10 |
> | --------------------------------- | ----------------- | ----------------- | -------------- | -------------- |
> | $\alpha$=0.5 (used in this study) | 67.80             | 70.40             | 68.84          | 71.54          |
> | $\alpha$=0.25                     | 68.45             | 70.63             | 69.22          | 71.84          |
> | $\alpha$=0.75                     | 67.21             | 68.54             | 68.64          | 70.28          |
>
> The results show that a smaller $\alpha$ may result in better performance.
>
>
>
> To further address the reviewer's concern, we have revised the result in Table 3, reporting the mean and std over three runs with different initializations.
>
> | Stable Learner | Plastic Learner | iCaRL              | WA                 | DER                | Foster             | MEMO               | Average       |
> | -------------- | --------------- | ------------------ | ------------------ | ------------------ | ------------------ | ------------------ | ------------- |
> | Sta-Net        | Pla-Net         | **70.21$\pm$0.19** | **71.53$\pm$0.13** | **75.26$\pm$0.20** | **73.18$\pm$0.04** | **74.44$\pm$0.07** | **72.92**     |
> | Sta-Net        | None            | 66.69$\pm$0.10     | 69.33$\pm$0.17     | 72.47$\pm$0.07     | 70.84$\pm$0.24     | 72.11$\pm$0.27     | 70.29 (-2.63) |
> | Pla-Net        | Pla-Net         | 69.57$\pm$0.10     | 69.98$\pm$0.08     | 74.64$\pm$0.28     | 71.92$\pm$0.28     | 69.80$\pm$0.15     | 71.18 (-1.74) |
> | Sta-Net        | Sta-Net         | 69.27$\pm$0.26     | 71.38$\pm$0.25     | 74.30$\pm$0.08     | 72.65$\pm$0.11     | 73.77$\pm$0.05     | 72.27 (-0.65) |
> | Pla-Net        | Sta-Net         | 69.85$\pm$0.13     | 70.31$\pm$0.26     | 74.25$\pm$0.35     | 71.86$\pm$0.06     | 69.92$\pm$0.12     | 71.24 (-1.68) |
>
> Moreover, we note that we have evaluated our proposed framework by combining it with a lot of existing CL methods to fully validate its effectiveness. For instance, if we employ the Wilcoxon Signed Rank Test to calculate the confidence level at which Dual-Arch significantly outperforms using two Sta-Net across CL methods, the results yield a p-value of 0.03125, which is less than the significance level of 0.05. This indicates that Dual-Arch is significantly better than using two Sta-Nets at a confidence level of 95%.
>
> For reference, the Wilcoxon Signed Rank Test can be implemented in R using the command:
>
> wilcox.test(c(70.21 ,71.53 ,75.26 ,73.18 ,74.44), c(69.27 ,71.38 ,74.30 ,72.65 ,73.77), paired=TRUE, alternative = "greater")

---

> > ### Author Response · Authors · 2024-11-23
> > **Initial Response -- part 3**
> >
> > >Weakness3: Comparison with related works.
> >
> > Thank you for your valuable comment. We note that the unique contributions of this study lie in our proposed method and its divergent research perspective compared to most existing methods, as highlighted in the Introduction.  Furthermore, our proposed framework has demonstrated the capability to be integrated with existing classical CL methods using distillation (e.g., iCarl and WA) or network expansion (e.g., DER and MEMO). Specifically, regarding the works you provided:
> >
> > -  "Heterogeneous Continual Learning" employs knowledge distillation to transfer knowledge from the previous model to the current one, whereas our work focuses on transferring knowledge from the current plastic learner to the current stable learner. These two transfer strategies can be combined in principle.  For example, we have already reported the result of integrating our framework with similar approaches (i.e., WA and iCaRL).
> > -  "Lifelong Learning with Dynamically Expandable Networks" allocates an incremental parameter space of the network for each new task, thereby mitigating the conflict between stability and plasticity at the parameter level. In this type of approach, performance can still be enhanced by modifying the initial architecture and introducing a plastic learner (see results of DER and MEMO).
> > -  "Decoupling Learning and Remembering"  stores the model information (i.e.,  projected knowledge) for the old tasks to mitigate forgetting, but does not focus on the impact of architectures, thus differing in research perspective from our work.
> >
> > To better highlight the unique contributions of our study, we have included a discussion that details the differences between these methods and our approach in the Related Works Section.
> >
> > >Question1: Details about KD
> >
> > Yes, we use the current data for knowledge distillation, as only the current data is available in CL. We have highlighted this in this revision to enhance clarity.
> >
> > >Question2: Results of forgetting.
> >
> > Thank you for your suggestion. We would like to clarify that the two metrics reported in Table 2—Last Accuracy (LA) and Average Incremental Accuracy (AIA)—are calculated with consideration of forgetting [2] [3]. Additionally, we are pleased to report the Final Average Forgetting (FAF) results (%) here, and we have included these results in this revision. Please note that parts of the results for ArchCraft in Table 2 are taken from [4], and since they do not provide FAF, we are unable to include their results here.
> >
> > | Method              | CIFAR100/20 | CIFAR100/10 | ImageNet100/20 | ImageNet100/10 |
> > | ------------------- | ----------- | ----------- | -------------- | -------------- |
> > | iCaRL               | 33.33       | 27.76       | 41.05          | 35.91          |
> > | w/ Dual-Arch (ours) | **29.49**   | **23.63**   | **35.66**      | **28.22**      |
> > | WA                  | **19.05**   | 23.53       | 39.05          | 28.27          |
> > | w/ Dual-Arch (ours) | 24.91       | **17.91**   | **31.73**      | **24.53**      |
> > | DER                 | 25.63       | 22.13       | 20.51          | 15.22          |
> > | w/ Dual-Arch (ours) | **20.08**   | **17.73**   | **16.97**      | **12.96**      |
> > | Foster              | 35.03       | 27.39       | 34.67          | 25.18          |
> > | w/ Dual-Arch (ours) | **23.75**   | **18.23**   | **32.42**      | **25.04**      |
> > | MEMO                | 35.71       | 27.99       | 29.64          | 21.87          |
> > | w/ Dual-Arch (ours) | **24.09**   | **20.82**   | **24.59**      | **16.53**      |
> >
> > >Question3: Details about training architectures in the ablation studies.
> >
> > The ‘Sta-Net’ and ‘Pla-Net’ denote the designed architectures and not directly determine the training loss.
> >
> > For Dual-Arch, the training loss for each network is determined by its role (i.e., stable learner or plastic learner), as detailed in Section 4.3. For instance, if we use two Pla-Net for both stable learner and plastic learner, the one acting as the plastic learner is trained solely using cross-entropy loss, while the one acting as the stable learner is trained using loss defined in Eq. (2), now presented as Eq. (4) in the current revision. Note that these training settings are consistent across all experiments with dual architectures in Table 3.
> >
> > For learning with a single learner, the network is trained using original CL methods, i.e., $L_{single} = L_{CE} +L_{CL}$, where $L_{CE}$ represents the cross-entropy loss, and $L_{CL}$ is specifically defined by the particular CL methods (e.g. iCaRL, WA).  We have added detailed information about this in Section 4.3 of this revision to enhance clarity.
> >
> > [1] Hinton et al. Distilling the Knowledge in a Neural Network.
> >
> > [2] Zhou et al. Pycil: A python toolbox for class-incremental learning.
> >
> > [3] Wang et al. A comprehensive survey of continual learning: theory, method and application.
> >
> > [4] Lu et al. Revisiting Neural Networks for Continual Learning: An Architectural Perspective.

---

> > > ### Comment · Reviewer_e7KZ · 2024-11-24
> > > **Thank you for your response**
> > >
> > > Thank you to the authors for their response. However, most of my concerns remain unresolved, and I keep my score. Below are my detailed comments:
> > >
> > > ---
> > >
> > > > Presentation of Table 1:
> > >
> > > The updated text could benefit from a clearer reflection in terms of parameters, as the motivation behind the table is still not evident. Additionally, the broader implications of these results on other architectures, as highlighted by other reviewers, remain unclear and need further explanation.
> > >
> > > ---
> > >
> > > > Details on Hyperparameters:
> > >
> > > From the additional experiments provided, it is evident that varying $\alpha$ impacts performance. This necessitates a more thorough analysis rather than relying on values from prior work, given the differences in context and methodology. Moreover, the reported mean and standard deviation suggest that the performance differences between the two sta-Net architectures, as well as between sta-Net and pla-Net, are less than 1%. This small margin underscores the need for further experiments on additional datasets to validate the conclusiveness of the results.
> > >
> > > ---
> > >
> > > > Comparison with Related Works:
> > >
> > > The comparison with Heterogeneous Continual Learning was intended to evaluate the knowledge distillation adoption, which is central to the proposed method. While DER and MEMO are replay-based methods, the requested comparison with expansion-based methods was relevant as they share similarities with the proposed approach regarding the inclusion of additional parameters.
> > >
> > > ---
> > >
> > > In summary, while the revisions addressed some aspects of the paper, more clarity is needed regarding the motivation, loss function, and hyper-parameter choices. Furthermore, the necessity of both STA-Net and PLA-Net architectures should be justified in future iterations.

---

> > > > ### Author Response · Authors · 2024-11-25
> > > > **Additional response**
> > > >
> > > > Thank you for your response. We hope the following additional clarification addresses your concerns, particularly regarding potential misunderstandings (for Weakness 3).
> > > >
> > > > > Weakness1: Presentation of Table 1:
> > > >
> > > > We have revised the description of Table 1 in Section 3.2 of this revision to more clearly articulate the motivation behind the analysis. Here is the updated content:
> > > >
> > > > "The performance comparison between ResNet-18 and its variants, under comparable parameter counts (within a $\pm$3\% margin), is summarized in Table 1. It can be observed that the wider yet shallower variant demonstrates decreases in AAN by 2.97% (83.44% vs. 86.41%) and FAF by 2.60% (33.16% vs. 35.76%), indicating enhanced stability but diminished plasticity. Similarly, modifying the penultimate layer to increase pre-classification width yields consistent results, with AAN decreasing by 1.77% (84.64% vs. 86.41%) and FAF by 1.59% (34.17% vs. 35.76%). These observations suggest that, within a fixed parameter budget, increasing network width may enhance stability at the expense of plasticity. Conversely, the two deeper yet narrower variants exhibit slight increases in AAN, rising by 0.27% and 0.43% (86.68% and 86.87% vs. 86.41%), as well as in FAF, with increases of 0.26% and 0.22% (36.02% and 35.98% vs. 35.76%), reflecting a slight trade-off favoring plasticity over stability.
> > > > This suggests that depth has a greater influence than width in plasticity under a given parameter constraint. Overall, the results reveal an inherent trade-off between stability and plasticity at the architectural level, governed by architectural design choices within specific parameter limits."
> > > >
> > > > To further address the reviewer's concern, we have conducted a comparative analysis of SepViT with varying depths and widths on ImageNet100/10. Here, "Depth" denotes the number of blocks, and "Width" denotes the dimension of the attention heads. Specifically, "Depth-11---Width-32" represents the original SepViT-Lite configuration.
> > > >
> > > > | Depth | Width | #Params | AAN↑          | FAF↓          |
> > > > | ----- | ----- | ------- | ------------- | ------------- |
> > > > | 11    | 32    | 3.78    | 79.54         | 40.51         |
> > > > | 5     | 49    | 3.76    | 78.96 (-0.58) | 39.47 (-1.04) |
> > > >
> > > > We observe that under comparable parameter counts, the wider yet shallower variant demonstrates decreases in AAN by 0.78% and FAF by 0.87%, indicating enhanced stability at the expense of plasticity. This trend mirrors that of Table 1 in our paper, thereby supporting our claim.
> > > >
> > > > >Weakness2: Details on Hyperparameters:
> > > >
> > > > While the study of hyperparameters is not the primary focus of our paper, we are glad to provide a more detailed analysis on this aspect. Specifically, we present the comparison between Sta-Net and Pla-Net (our method) on CIFAR100/10 using $\alpha=0.25$.
> > > >
> > > > | Stable Learner | Plastic Learner | iCaRL     | WA        | DER       | Foster    | MEMO      | Average       |
> > > > | -------------- | --------------- | --------- | --------- | --------- | --------- | --------- | ------------- |
> > > > | Sta-Net (ours) | Pla-Net (ours)  | **70.63** | **71.84** | **75.12** | **73.56** | **74.61** | **73.15**     |
> > > > | Sta-Net        | Sta-Net         | 69.83     | 71.51     | 73.45     | 73.24     | 73.58     | 72.32 (-0.83) |
> > > >
> > > > These results shows that although the hyperparameter $\alpha$ does influence the performance, it does not significantly alter the relative performance differences between our method and the baseline.
> > > >
> > > > To further address the reviewer's concern, we report the comparison on ImageNet100/10 using the default value of $\alpha$.
> > > >
> > > > | Stable Learner | Plastic Learner | iCaRL     | WA        | DER       | Foster    | MEMO      | Average       |
> > > > | -------------- | --------------- | --------- | --------- | --------- | --------- | --------- | ------------- |
> > > > | Sta-Net (ours) | Pla-Net (ours)  | **69.37** | **72.57** | **77.49** | **72.42** | **75.54** | **73.48**     |
> > > > | Sta-Net        | Sta-Net         | 68.84     | 71.59     | 75.55     | 72.14     | 74.57     | 72.54 (-0.94) |
> > > >
> > > > These results shows that our method consistently outperform the baseline on ImageNet100/10, demonstrating the importance of using dedicated architectures, one of key components within Dual-Arch.

---

> > > > > ### Author Response · Authors · 2024-11-25
> > > > > **Additional response -- part 2**
> > > > >
> > > > > >Weakness3: Comparison with Related Works (knowledge distillation-based and expansion-based):
> > > > >
> > > > > It should be noted that **we did include evaluations comparing our proposed framework both knowledge distillation-based methods (iCaRL and WA) and expansion-based methods (DER and MEMO) in our initial submission.** We reiterate that iCaRL and WA are representative knowledge distillation-based methods, DER and MEMO are representative expansion-base methods, as we have previously mentioned in the Related Works section and our initial response.
> > > > >
> > > > > To further address the reviewer's concerns, we have implemented "Heterogeneous Continual Learning" with Dual-Arch. Below are the results on CIFAR100 (20 tasks following the original setting):
> > > > >
> > > > > | Methods                          | LA (%) in Class IL | LA (%) in Task IL |
> > > > > | -------------------------------- | ------------------ | ----------------- |
> > > > > | Heterogeneous Continual Learning | 16.77              | 87.13             |
> > > > > | w/ Dual-Arch (ours)              | **23.07 (+6.30)**  | **90.19 (+3.06)** |
> > > > >
> > > > >
> > > > >
> > > > > For reference, we provide a brief introduction to iCaRL, WA, DER, and MEMO, which are widely used as baselines in existing works, such as [1] [2].
> > > > >
> > > > > - iCaRL [3] builds knowledge distillation regularization term to regularize former classes from being forgotten.
> > > > > - WA [4] extends knowledge distillation-based CL with weight aligning, which normalizes the linear layers to reduce the negative bias.
> > > > > - DER [5] expands a new backbone when facing new tasks and aggregates the features with a larger linear layer.
> > > > > - MEMO [6] expands the specialized blocks of the backbone when facing new tasks.
> > > > >
> > > > >
> > > > >
> > > > > [1] Wang et al. Beef: Bi-compatible class-incremental learning via energy-based expansion and fusion.
> > > > >
> > > > > [2] Cha et al. Hyperparameters in Continual Learning: A Reality Check.
> > > > >
> > > > > [3] Rebuffi et al. icarl: Incremental classifier and representation learning.
> > > > >
> > > > > [4] Zhao et al. Maintaining discrimination and fairness in class incremental learning.
> > > > >
> > > > > [5] Yan et al. Der: Dynamically expandable representation for class incremental learning.
> > > > >
> > > > > [6] Zhou et al. A model or 603 exemplars: Towards memory-efficient class-incremental learning.
> > > > >
> > > > >
> > > > > ------
> > > > >
> > > > > Thank you for your response again. We hope this response addresses your concerns and clarifies any misunderstandings regarding our study. We kindly request further discussion to address any specific concerns or questions you may have.

---

> > > > ### Author Response · Authors · 2024-12-02
> > > >
> > > > Dear Reviewer e7KZ,
> > > >
> > > > Hope you had a wonderful Thanksgiving holiday week! We would like to sincerely thank you once again for your valuable comments and your time spent on our work. We believe that all your questions can be well addressed in the responses and revised version of the paper.
> > > >
> > > > As the extended discussion period ends tomorrow, we would greatly appreciate your feedback soon. If our rebuttal adequately addresses your concerns, we kindly request an update on your evaluations. Once again, thank you very much for your reviews and feedback!
> > > >
> > > > Yours sincerely,
> > > >
> > > > Authors of Paper 6928

---

> ### Author Response · Authors · 2024-11-27
> **A Gentle Reminder of Feedbacks**
>
> Dear Reviewer e7KZ:
>
> Thanks for your careful comments and your time spent on our work. We have revised the paper accordingly and incorporated additional discussions and experiments addressing your concerns.
>
> Currently, all of your concerns can be resolved in the responses and revised version of the paper. We want to leave a gentle reminder that the revision deadline is approaching. We would appreciate your feedback to make sure that our responses and revisions have resolved your concerns, or whether there is a leftover concern that we can address to ensure the quality of our work.
>
> Yours sincerely,
>
> Authors of Paper 6928

---

### Official Review · Reviewer_skk8 · 2024-11-02

**Soundness:** 3
**Presentation:** 2
**Contribution:** 2
**Rating:** 6
**Confidence:** 3

**Summary:**

Building on the architectural design of continual learning (CL), the paper proposes that wide and shallow network layers exhibit better stability, while deep and thin networks show better plasticity. It combines these two structures using knowledge distillation, improving model performance under the same parameter count. The paper adopts specific model structures for different requirements, inspiring optimization from a fundamental architectural perspective rather than merely focusing on weight optimization. However, the experiments are all based on ResNet18, lacking validation on a wider range of model architectures.

**Strengths:**

This paper extends previous research by exploring the relationship between model width, depth, and model stability and plasticity. It innovatively uses knowledge distillation to combine two structures, demonstrating strong originality. Unlike most previous continual learning methods that focused on optimizing weights, this paper employs different architectures for different task requirements, inspiring optimization from a fundamental architectural perspective.

**Weaknesses:**

This paper does not sufficiently emphasize the limitations of previous methods and the advantages of its own approach. For instance, designing the framework based on structural characteristics. The value of the paper is not clearly highlighted. Additionally, the methods section lacks mathematical descriptions, such as the definition of continual learning and the specific loss function formulas. Moreover, the experiments are conducted only on ResNet18, lacking validation on a broader range of model architectures, such as transformers.

**Questions:**

1.  What is the workflow of the testing process
2.  How is the replay mechanism implemented?
3.  Should Table 1 indicate the original ResNet for easier reading by the audience?
4. Why is there no experiment involving Pla-Net---SatNet in the ablation study in Table 3?

---

> ### Author Response · Authors · 2024-11-23
> **Initial Response**
>
> We appreciate the reviewer's recognition of our innovative research perspective, findings, and approach. To further address the reviewer's concern, we have:
>
> - revised the Introduction and Related Works to better emphasize the limitations of previous methods and the advantages of our approach (for Weakness 1).
> - included detailed mathematical descriptions (for Weakness 1, Question 1,2).
> - revised Table 1 follows the suggestion (for Question 3).
> - included additional ablation experiments (for Question 4).
>
> > Weakness1: Better presentation of this study's value.
>
> Thank you for your valuable comments. While we have highlighted the unique contributions of our study by emphasizing its distinct research perspective, we acknowledge the need to further emphasize the limitations of previous methods and the advantages of our approach. To address this, we have revised the related content in the Introduction Section and expanded the discussion in the Related Works Section to clearly delineate the advantages of our approach beyond existing methods.
>
> > Weakness2: Mathematical Descriptions.
>
> We appreciate the reviewer's suggestion for detailed mathematical descriptions. In this revision, we have incorporated these descriptions in Section 4.3, including the definitions of continual learning, the cross-entropy loss ($L_{CE}$), and the knowledge distillation loss ($L_{KD}$). Additionally, we would like to clarify that the continual learning loss ($L_{CL}$) is an abstract term that is specifically designed by the continual learning methods in combination with our approach.
>
> > Weakness3: Experiments on other architectures
>
> While our study primarily focuses on ResNet, the insights presented in our paper could extend to other architectures, such as Vision Transformers (ViTs). Here, we report the results of transferring our method to SepViT [1] on ImageNet100/10. The results demonstrate that Dual-Arch consistently enhances the CL performance of SepViT, indicating its generality to ViTs. We have also included these results in Section A.7 of this revision.
>
> | Method              | Params (M) | LA (%)            | AIA (%)           |
> | ------------------- | ---------- | ----------------- | ----------------- |
> | iCaRL               | 7.57       | 43.08             | 60.62             |
> | w/ Dual-Arch (ours) | 5.32       | **46.34 (+3.26)** | **63.09 (+2.47)** |
> | WA                  | 7.57       | 38.40             | 57.67             |
> | w/ Dual-Arch (ours) | 5.32       | **44.28 (+5.88)** | **61.15 (+3.48)** |
>
> >Question1: Workflow of the testing process.
>
> Following [2], after the learning of each task, the (main) model is evaluated on the test data of the current task as well as all previously learned tasks. We have detailed this content in Section 4.3 of this revision.
>
> > Question2: Implementation of replay.
>
> Following [2] [3], to implement replay, parts of data from previous tasks are preserved and subsequently incorporated into the training data for all models. We have detailed this content in Section 4.3 of this revision.
>
> > Question3: Indicating the original ResNet in Table 1.
>
> We appreciate the suggestion and have highlighted this aspect to enhance readability in this revision.
>
> >Question4: Experiment involving Pla-Net---StaNet.
>
> We have incorporated such an experiment in this revision, and the results are as follows.
>
> | Stable Learner | Plastic Learner | iCaRL          | WA             | DER            | Foster         | MEMO           | Average       |
> | -------------- | --------------- | -------------- | -------------- | -------------- | -------------- | -------------- | ------------- |
> | Sta-Net        | Pla-Net         | **70.21$\pm$0.19** | **71.53$\pm$0.13** | **75.26$\pm$0.20** | **73.18$\pm$0.04** | **74.44$\pm$0.07** | **72.92**         |
> | Sta-Net        | None            | 66.69$\pm$0.10 | 69.33$\pm$0.17 | 72.47$\pm$0.07 | 70.84$\pm$0.24 | 72.11$\pm$0.27 | 70.29 (-2.63) |
> | Pla-Net        | Pla-Net         | 69.57$\pm$0.10 | 69.98$\pm$0.08 | 74.64$\pm$0.28 | 71.92$\pm$0.28 | 69.80$\pm$0.15 | 71.18 (-1.74) |
> | Sta-Net        | Sta-Net         | 69.27$\pm$0.26 | 71.38$\pm$0.25 | 74.30$\pm$0.08 | 72.65$\pm$0.11 | 73.77$\pm$0.05 | 72.27 (-0.65) |
> | Pla-Net        | Sta-Net         | 69.85$\pm$0.13 | 70.31$\pm$0.26 | 74.25$\pm$0.35 | 71.86$\pm$0.06 | 69.92$\pm$0.12 | 71.24 (-1.68) |
>
> The configuration 'Pla-Net---StaNet' demonstrates inferior performance compared to our original setup, 'Sta-Net---PlaNet'. Additionally, it marginally outperforms 'Pla-Net---Pla-Net', which may be attributed to the potential benefits derived from the integration of different architectures.
>
>
> [1] Li, et al. Sepvit: Separable vision transformer.
>
> [2] Zhou et al. Pycil: A python toolbox for class-incremental learning.
>
> [3] Rebuffi et al. icarl: Incremental classifier and representation learning.

---

> > ### Comment · Reviewer_skk8 · 2024-11-26
> >
> > Thanks for the rebutall. This rebutall adresses most of my concerns. I will increase my score.

---

> > > ### Author Response · Authors · 2024-11-27
> > >
> > > Thank you for your valuable comments and for acknowledging our work.
> > >
> > > Your valuable suggestions significantly improved the quality of our manuscripts. We appreciate your time and effort in reviewing our work. Thanks!

---

### Official Review · Reviewer_ULta · 2024-11-03

**Soundness:** 2
**Presentation:** 3
**Contribution:** 2
**Rating:** 5
**Confidence:** 2

**Summary:**

In this paper, the authors demonstrate the impact of architecture of a neural network to its capability in continual learning (CL). They show that with same amount of parameters, a wider network has a higher stability and a deeper network has a higher plasticity. From this observation, they develop a CL framework featuring two networks, each with architecture adjusted to optimize one of the stability and plasticity. They showed through experiments that this framework could be used in combination with several popular CL methods to achieve enhanced performance and reduced parameter counts.

**Strengths:**

The basic idea of the proposed method is based on a sound and clear observation about the relation between a model's architecture and its stability/plasticity characteristics in CL tasks.

Additional experiments including ablation studies and analysis of the stability-plasticity tradeoff make a strong case.

**Weaknesses:**

Plasticity and stability are not static properties of an architecture. For example, throughout the continual learning process, if a model is trained more for each task, the model will appear to be more plastic. This simple observation weakens the argument of a “stability-plasticity” tradeoff at the architecture level.

Most experiments use CIFAR-100 for which the ResNet-18 architecture might be over-parameterized. This raises questions about the validity of the claimed performance and parameter efficiency improvements.

**Questions:**

The paper focuses on specific variants of ResNet, which exemplify the stability-plasticity tradeoff. It would be interesting to see if this tradeoff generalizes to other variants of ResNet or even other architectures.

If we compare Table 2 and Table 3, we can see that the performance of ResNet18 without Dual-Arch is comparable or worse than the performance of using “Sta-Net” as a stable learner only. Thus, “Sta-Net” seems to be a more reasonable choice of baselines, especially if previous works already demonstrated that “wider and shallower networks exhibit superior overall CL performance”.

While it is likely true that Dual-Arch could improve parameter efficiency, the specific cases shown in the paper may exaggerate the differences. As in Figure 3a, performance of ResNet-18 exhibits very limited performance drop with reduction in number of parameters. This suggests that ResNet-18 is already over-parameterized for the CIFAR-100/10 task.

In the design of the Dual-Arch, only the stable learner is involved in the prediction and evaluation. What is the mechanism for this stable learner in Dual-Arch to exhibit plasticity similar to or even more than the plasticity of a single plastic learner (as shown in Figure 4c).
This is intriguing because structurally, the stable learner should be less plastic; and during training, both models received the same epochs of training. It would be nice if the authors could provide some explanation.

One explanation would be that a stable learner is able to learn more from the data because it gains advantage through learning from the distilled knowledge. It seems fair to also compare the proposed method with a stable learner with twice the amount of training epochs - so we can control the total training to be same.

---

> ### Author Response · Authors · 2024-11-23
> **Initial Response**
>
> We appreciate the reviewer's acknowledgment of our sound observation and idea, as well as detailed experiments and analysis. To further address the reviewer's concern, we have:
>
> - provided further clarification and experiment to support our argument of a stability-plasticity tradeoff at the architecture level (for Weakness 1).
> - explained why Resnet18 is selected as the baseline (for Weakness 2, Question 2, 3)
> - included a further experimental evaluation on ViTs (for Question 1).
> - provided an explanation of the enhanced plasticity within our proposed framework (for Question 4).
>
> > Weakness1: Concern about the stability-plasticity tradeoff at the architecture level.
>
> We acknowledge that stability and plasticity can vary in different training environments, which is due to the impact of model parameters on the stability-plasticity tradeoff. However, it is crucial to emphasize that the architecture itself also inherently influences these properties. For instance, under the same training settings, ResNet typically exhibits greater plasticity compared to a deep MLP in most scenarios. Our study, conducted under consistent training settings, reveals that certain architectures are inherently more plastic, while others are more stable. This suggests that the architecture plays a fundamental role in stability and plasticity. We believe that while training settings and model parameters (i.e., weights) are important, they do not diminish the existence of the stability-plasticity tradeoff at the architecture level. Instead, they highlight the need for considering both architectural design and learning methods to achieve optimal performance.
>
> Moreover, we note that the experimental results in our responses to Question 4 also support our argument of a stability-plasticity tradeoff at the architecture level.
>
> > Weakness2: Concern about the using of ResNet18 on CIFAR-100.
>
> We appreciate the reviewer's valuable feedback. However, we believe it is not entirely accurate to conclude that ResNet-18 is over-parameterized for CIFAR100 in the context of Continual Learning (CL). First, ResNet-18 is widely adopted by many existing CL methods and benchmarks for the CIFAR100 dataset, including [1-4]. These existing works support the use of ResNet-18 for CIFAR-100 in CL, indicating that its parameter counts are suitable for achieving competitive performance. Moreover, expansion-based methods that continually expand network parameters often outperform methods with fixed parameters when using ResNet-18 as the backbone in CL. For instance, DER achieves significantly superior CL performance compared to WA and Foster on CIFAR-100, as demonstrated in Table 2 of our paper. This observation further suggests that ResNet-18 might not be over-parameterized for CIFAR-100 in CL.
>
> Moreover, regarding Question 2, it is important to clarify that the limited performance drop in Figure 3a might be due to the significant increase in parameter counts introduced by DER, which expands from 11.2M to 112M parameters. This substantial parameter expansion sightly masks the inherent parameter efficiency of ResNet-18. To further illustrate this point, we can observe from Figure 3b of our paper that when combining ResNet-18 with Foster, an expansion-free method, the performance drops considerably with a reduction in parameter counts. This indicates that ResNet-18 is not inherently over-parameterized for the CIFAR-100/10 task. We believe that reporting the results of both representative expansion-based and expansion-free methods makes our parameter efficiency analysis comprehensive.
>
> [1] Yan et al. Der: Dynamically expandable representation for class incremental learning.
>
> [2] Arani et al. Learning fast, learning slow: A general continual learning method based on complementary learning system.
>
> [3] Madaan et al. Heterogeneous Continual Learning.
>
> [4] Sun et al. Decoupling Learning and Remembering: a Bilevel Memory Framework with Knowledge Projection for Task-Incremental Learning.
>
> > Question1: Experiments on other architectures.
>
> While our study primarily focuses on ResNet, the insights presented in our paper could extend to other architectures, such as Vision Transformers (ViTs). Here, we report the results of transferring our method to SepViT [5] on ImageNet100/10. The results demonstrate that Dual-Arch consistently enhances the CL performance of SepViT, indicating its generality to ViTs. We have also included these results in Section A.7 of this revision.
>
> | Method              | Params (M) | LA (%)            | AIA (%)           |
> | ------------------- | ---------- | ----------------- | ----------------- |
> | iCaRL               | 7.57       | 43.08             | 60.62             |
> | w/ Dual-Arch (ours) | 5.32       | **46.34 (+3.26)** | **63.09 (+2.47)** |
> | WA                  | 7.57       | 38.40             | 57.67             |
> | w/ Dual-Arch (ours) | 5.32       | **44.28 (+5.88)** | **61.15 (+3.48)** |
>
> [5] Li, et al. Sepvit: Separable vision transformer.

---

> > ### Author Response · Authors · 2024-11-23
> > **Initial Response -- part 2**
> >
> > > Question2: Concern about baselines.
> >
> > We acknowledge the importance of comparing our proposed framework with architectures that exhibit superior overall CL performance. This is why we included ArchCraft as an additional baseline in our initial submission, which employs a single, CL-friendly architecture, as shown in Table 2 of our paper. However, we still believe that employing ResNet-18 as a baseline is necessary and better than using Sta-Net for two primary reasons:
> >
> > (1) ResNet-18 is a widely adopted architecture in existing CL methods, as mentioned above, making it a standard baseline for comparison. Its widespread use in CL ensures that our results are directly comparable to a broad range of previous studies.
> >
> > (2) Both architectures used in Dual-Arch (i.e., Sta-Net and Pla-Net) are derived from modifications to ResNet-18. Therefore, it is meaningful to conduct a comparison between ResNet-18 and our proposed framework.
> >
> > By including both ResNet-18 and ArchCraft as baselines, we aim to provide a comprehensive evaluation that encompasses both widely used architectures and those specifically designed for superior CL performance.
> >
> > >Question3: Concern about the using of ResNet18 on CIFAR-100.
> >
> > Please see our response to Weakness 2
> >
> > > Question4: Concern about the superior plasticity of Dual-Arch compared to Pla-Net.
> >
> > We appreciate the reviewer's insightful question regarding the mechanism by which the stable learner in Dual-Arch exhibits plasticity similar to or even greater than that of a single plastic network, as shown in Figure 4c. We believe that the explanation provided by the reviewer—that the stable learner gains plasticity through learning from the distilled knowledge of the plastic learner—is indeed accurate. Here's a more detailed explanation of how our proposed framework operates: The stable learner, equipped with a stable architecture, is inherently stable. Moreover, its plasticity can be enhanced by learning the distilled knowledge from the plastic learner, which is equipped with a more plastic architecture. This Dual-Arch mechanism allows the stable learner to benefit from the plastic learner's plasticity without compromising its inherent stability.
> >
> > To address the reviewer's concern more comprehensively, we have conducted additional experiments with single learners trained for twice the number of epochs.
> >
> > | Architecture     | Training Epochs (double for the first task) | AAN (%)   |
> > | ---------------- | ------------------------------------------- | --------- |
> > | Dual-Arch (ours) | 100                                         | **81.47** |
> > | Sta-Net          | 100                                         | 67.58     |
> > | Sta-Net          | 200                                         | 69.23     |
> > | Pla-Net          | 100                                         | 78.75     |
> > | Pla-Net          | 200                                         | 80.53     |
> >
> > These results demonstrate that our Dual-Arch method still outperforms using a single Pla-Net, even when the latter receives extended training. This further validates the effectiveness of our approach in balancing stability and plasticity at the architectural level. Moreover, we can observe that the Pla-Net consistently exhibits superior plasticity compared to Sta-Net, which further supports our argument regarding the inherent stability-plasticity tradeoff at the architecture level.

---

> > ### Comment · Reviewer_ULta · 2024-11-24
> >
> > I appreciate the extra effort invested by the authors in responding to the reviews and revising the paper. However, I am still concerned about the validity of the main premise ("cornerstone") in which this entire work is based on -- namely that deeper networks exhibit greater plasticity and wider networks exhibit greater stability.
> >
> > Also, after reading the rest of the reviews, I feel more confident that this paper is not ready for publication.

---

> > > ### Author Response · Authors · 2024-11-25
> > > **Additional response**
> > >
> > > Thank you for your response. We hope the following additional clarification addresses your concerns.
> > >
> > > > Concern about the validity of the stability-plasticity tradeoff at the architecture level.
> > >
> > > We acknowledge that our methods are based on findings that within a fixed parameter budget, certain architectural designs (wide and shallow) are inherently stable, while others (deep and thin) are plastic. We have conducted various experiments on ResNet and MLP, as shown in Section 3 and A.4, to support these findings. Moreover, these conclusions are partially (i.e., wide and shallow --> stability) corroborated by existing works, such as [1] [2] [3], which we mentioned in the Introduction and Related Works sections.
> > >
> > > To further address the reviewer's concern, we conducted a comparative analysis of ViT (specifically, SepViT) with varying depths and widths on ImageNet100/10.
> > >
> > > | Depth | Width | #Params | AAN↑          | FAF↓          |
> > > | ----- | ----- | ------- | ------------- | ------------- |
> > > | 11    | 32    | 3.78    | 79.54         | 40.51         |
> > > | 5     | 49    | 3.76    | 78.96 (-0.58) | 39.47 (-1.04) |
> > >
> > > It can be observed that under comparable parameter counts and the same training setting, the wider yet shallower variant demonstrates decreases in AAN by 0.78% and FAF by 0.87%, indicating enhanced stability at the expense of plasticity. This trend mirrors that of Table 1 and Table 4 in our paper, thereby supporting our claim.
> > >
> > > Additionally, in the initial comments, the reviewer highlighted that one of the strengths of our paper is "The basic idea of the proposed method is based on a sound and clear observation about the relation between a model's architecture and its stability/plasticity characteristics in CL tasks." We would be grateful if the reviewer could provide further insight into what may have contributed to the change in this opinion.
> > >
> > >
> > >
> > > [1] Mirzadeh et al. Wide neural networks forget less catastrophically.
> > >
> > > [2] Mirzadeh et al. Architecture matters in continual learning.
> > >
> > > [3] Lu et al. Revisiting neural networks for continual learning: An architectural perspective.
> > >
> > > ------
> > >
> > > Thank you for your response again. We hope that this response addresses your concern. We kindly request further discussion to address specific reasons behind your concern or any other questions you may have.

---

> > > ### Author Response · Authors · 2024-11-27
> > > **A Gentle Reminder of Feedbacks**
> > >
> > > Dear Reviewer ULta:
> > >
> > > Thanks for your careful comments and your time spent on our work. We have revised the paper accordingly and incorporated additional discussions and experiments addressing your concerns.
> > >
> > > Currently, all of your concerns can be resolved in the responses and revised version of the paper. We want to leave a gentle reminder that the revision deadline is approaching. We would appreciate your feedback to make sure that our responses and revisions have resolved your concerns, or whether there is a leftover concern that we can address to ensure the quality of our work.
> > >
> > > Yours sincerely,
> > >
> > > Authors of Paper 6928

---

> > > ### Author Response · Authors · 2024-12-02
> > >
> > > Dear Reviewer ULta,
> > >
> > > Hope you had a wonderful Thanksgiving holiday week! We would like to sincerely thank you once again for your valuable comments and your time spent on our work. We believe that all your questions can be well addressed in the responses and revised version of the paper.
> > >
> > > As the extended discussion period ends tomorrow, we would greatly appreciate your feedback soon. If our rebuttal adequately addresses your concerns, we kindly request an update on your evaluations. Once again, thank you very much for your reviews and feedback!
> > >
> > > Yours sincerely,
> > >
> > > Authors of Paper 6928

---

### Official Review · Reviewer_uYip · 2024-11-03

**Soundness:** 2
**Presentation:** 3
**Contribution:** 2
**Rating:** 6
**Confidence:** 3

**Summary:**

This paper studies the stability-plasticity trade-off in Continual Learning from an architectural perspective. Its main finding is that there is a trade-off between the deep and the wide property of neural networks. In principle, the deeper networks cope better with plasticity, while wider networks cope better with stability. Furthermore, the paper also proposes a framework which makes use of distillation to improve continual learning performance on top of several baseline methods.

**Strengths:**

S1) The main finding (claim) (deeper networks help with plasticity, while wider networks help with stability) is, up to my best knowledge, novel and interesting.

S2) The proposed dual architecture seems to offer a fair increase in performance.

S3) Well written and easy to understand paper.

S4) The code is provided for easy reproducibility.

**Weaknesses:**

W1) The paper has a relatively low level of novelty as it mainly combines existing observations and techniques. Although, it seems to do all these from a novel perspective.

W2) Empirical evaluation of the main claim could have been more thorough.

W3) The paper needs a careful proofread. For instance, there are some typos: e.g., Tables 1 and 4 Captions “ANN”->”AAN”.

**Questions:**

Q1) Your main finding from Section 3 (deep->plasticity; wide->stability) was tested just in combination with iCaRL if my understanding is correct. This may throw some shadows on its generality. Have you considered evaluating everything in combination also with other baseline methods?

Q2) Have you considered more realistic settings with many more tasks? (considerably higher than 20)

Q3) (optional) The proposed dual architecture framework shows promise for class-incremental learning. Have you tested or considered its effectiveness also in other continual learning settings?

---

> ### Author Response · Authors · 2024-11-23
> **Initial Response**
>
> We appreciate the reviewer's recognition of our novel and interesting findings, the fair performance improvements, and the clarity of our paper. To further address the reviewer's concern, we have:
>
> - provided further clarification to highlight the unique contribution of our work (for Weakness 1).
> - included further experimental evaluation with additional baseline, incremental step, continual learning scenario (for Weakness 2, Question 1, 2, 3).
> - proofread the paper carefully to correct typos (for Weakness 3).
>
>  > Weakness1: Level of novelty.
>
> While existing observations and techniques have indeed motivated this work, our study offers unique contributions beyond merely combining them. Specifically, this paper is the first to explicitly identify that a stable architecture limits plasticity, while a plastic one limits stability, reflecting the stability-plasticity trade-off at the architectural level within continual learning (CL).  Moreover, this novel issue is addressed through our innovative approach of combining two distinct and specialized architectures, which has not been explored in prior CL works, to the best of our knowledge. This exploration and method innovation distinguish our work and contribute to the advancement of CL research.
>
> We have also revised the Introduction to highlight the unique contribution of our work. We hope this clarification underscores the distinctiveness of our contributions and the value of our intuitive yet novel approach.
>
> > Weakness2: More thorough evaluation.
>
> Please see our responses to the Questions.
>
> > Weakness3: Proofread.
>
> We have corrected the typo, changing "ANN" to "AAN." We have thoroughly proofread the manuscript to address every other issue we found. Thank you for pointing this out.
>
> > Question1:Evaluation in combination with other methods
>
> We acknowledge the importance of a more thorough evaluation. To further address this concern, we have conducted an empirical evaluation in combination with ADC[1], a state-of-the-art replay-free CL method. The results on CIFAR100/10 are presented below.
>
> | Depth | Width | Penultimate Layer | AAN (%)*↑*    | FAF (%)*↓*    |
> | ----- | ----- | ----------------- | ------------- | ------------- |
> | 18    | 64    | GAP               | 68.46         | 31.43         |
> | 10    | 96    | GAP               | 65.79 (-2.67) | 27.02 (-4.41) |
> | 18    | 64    | 2 *×* 2 AvgPool   | 64.73 (-3.73) | 27.63 (-3.80) |
> | 26    | 52    | GAP               | 69.88 (+1.42) | 32.93 (+1.50) |
> | 34    | 46    | GAP               | 69.55 (+1.09) | 32.82 (+1.39) |
>
> These results also suggest that, in comparison with existing architectural designs, the wider and shallower variants may offer superior stability at the cost of reduced plasticity.  This trend mirrors that of Table 3 in our paper, thereby supporting our claim.
>
> > Question2:Experiments on settings with more tasks.
>
> We are pleased to report the results on settings with a greater number of tasks (e.g., 50). Below are the results on CIFAR100/50. These results show that our method still outperforms the baselines in this more realistic setting. We have also included these results in Section A.5 of this revision.
>
> | Method              | LA (%)    | AIA (%)   |
> | ------------------- | --------- | --------- |
> | iCaRL               | 45.30     | 63.99     |
> | w/ ArchCraft        | 48.70     | **67.24** |
> | w/ Dual-Arch (ours) | **48.95** | 65.93     |
> | WA                  | 42.12     | 58.26     |
> | w/ ArchCraft        | 39.83     | 61.02     |
> | w/ Dual-Arch (ours) | **47.13** | **64.41** |
> | DER                 | 55.73     | 69.53     |
> | w/ ArchCraft        | 57.89     | 71.53     |
> | w/ Dual-Arch (ours) | **61.88** | **73.09** |
> | Foster              | 43.45     | 59.81     |
> | w/ ArchCraft        | 53.02     | **68.14** |
> | w/ Dual-Arch (ours) | **53.16** | 67.83     |
> | MEMO                | 42.44     | 62.57     |
> | w/ ArchCraft        | 54.47     | 69.96     |
> | w/ Dual-Arch (ours) | **58.09** | **71.17** |

---

> > ### Author Response · Authors · 2024-11-23
> > **Initial Response -- part 2**
> >
> > > Question3: Experiments in other continual learning settings
> >
> > While our primary focus is on class-incremental learning, our proposed framework has the potential to benefit other CL settings. This is because the core challenge that our proposed framework addresses—the stability-plasticity trade-off—is a common issue across various CL scenarios. To further explore this, we conducted experiments in a more challenging setting, general continual learning [2] [3], where task boundaries are not explicitly available. These results indicate that Dual-Arch consistently enhances CL performance in this scenario, underscoring its broad applicability. We have also included these results in Section A.6 of this revision.
> >
> > | Method              | Buffer Size 500   | Buffer Size 1000  |
> > | ------------------- | ----------------- | ----------------- |
> > | ER [4]              | 20.30             | 34.13             |
> > | w/ Dual-Arch (ours) | **27.57 (+7.27)** | **35.40 (+1.27)** |
> > | DER++ [3]           | 25.82             | 33.64             |
> > | w/ Dual-Arch (ours) | **30.34 (+4.52)** | **36.84 (+3.20)** |
> >
> >
> > [1] Goswami et al. Resurrecting Old Classes with New Data for Exemplar-Free Continual Learning.
> >
> > [2] Arani et al. Learning fast, learning slow: A general continual learning method based on complementary learning system.
> >
> > [3] Buzzega et al. Dark experience for general continual learning: a strong, simple baseline.
> >
> > [4] Rostam et al.  Complementary learning for overcoming catastrophic forgetting using experience replay.

---

> ### Author Response · Authors · 2024-11-27
> **A Gentle Reminder of Feedbacks**
>
> Dear Reviewer uYip:
>
> Thanks for your careful comments and your time for our work. We have revised our paper and added the discussion and experiments concerning.
>
> Currently, all of your concerns can be resolved in the responses and revised version of the paper. However, as the revision deadline is approaching, we kindly request your feedback to confirm that our responses and revisions have effectively resolved your concerns. If there are any remaining issues, we would be grateful for the opportunity to address them to ensure the quality of our work.
>
> Yours sincerely,
>
> Authors of Paper 6928

---

> > ### Comment · Reviewer_uYip · 2024-12-01
> > **Rebuttal acknowledged**
> >
> > I thank the authors for considering my comments and trying to address my concerns. After reading their answers, I have decided to increase my rating from 5 to 6.

---

> > > ### Author Response · Authors · 2024-12-01
> > >
> > > Thank you for your valuable comments and positive feedback.
> > >
> > > Your insightful comments have substantially enhanced the clarity and overall quality of our paper. We appreciate your time and effort in reviewing our work. Thanks!

---

### Meta-Review · Area_Chair_j99v · 2024-12-19

**Metareview:**

This paper addresses the challenge of stability/plasticity in continual learning. The authors propose a dual architecture in which one model preserves utility while the other acquires new knowledge, aiming to enhance the stability/plasticity balance. The approach is empirically validated through image classification tasks (e.g., using CIFAR and a subset of ImageNet datasets). While the study provides a comprehensive evaluation with ResNet18, it remains unclear whether the proposed solution generalizes to other architectures, such as transformers or more modern CNNs. Although the authors conducted an additional experiment with a separable vision transformer during the rebuttal, the reviewers were not fully convinced of the solution's generalizability. Additionally, concerns remain regarding the optimality of the stable and plastic learner designs in the current setting.

Given these concerns, the AC, sadly, recommends rejection. We hope this review helps the authors identify areas for improvement. Specifically, demonstrating generalization beyond the currently evaluated architecture and offering empirical justifications—such as why the stable learner cannot share a similar architecture with the plastic learner—would help strengthen the work and better establish its contributions.

**Additional Comments On Reviewer Discussion:**

The author-reviewer discussion period saw a productive dialogue where the authors addressed several reviewer concerns to their satisfaction. For example, this included new experiments using a separable vision transformer. Following this period, the AC initiated a discussion among the reviewers. Unfortunately, none of the reviewers championed the study, citing a lack of convincing arguments regarding the generalizability of the idea to other architectures and the absence of experiments or justification for the format of the plasticity learner (e.g., why not use two sta-nets with different learning dynamics?). Overall, while the work has its merits, the reviewers believe it lacks sufficient justification—both theoretical and empirical—to be considered a general solution for continual learning.

---

### Decision · Program_Chairs · 2025-01-22

Reject